# REPRESENTATION BALANCING
# OFFLINE MODEL-BASED REINFORCEMENT LEARNING

**Byung-Jun Lee**[1,3]**, Jongmin Lee**[1] **& Kee-Eung Kim**[1,2]
[1]School of Computing, KAIST, Daejeon, Republic of Korea
[2]Graduate School of AI, KAIST, Daejeon, Republic of Korea
[3]Gauss Labs Inc., Seoul, Republic of Korea
{bjlee,jmlee}@ai.kaist.ac.kr, kekim@kaist.ac.kr

## ABSTRACT

One of the main challenges in offline and off-policy reinforcement learning is to cope with the distribution shift that arises from the mismatch between the target policy and the data collection policy. In this paper, we focus on a model-based approach, particularly on learning the representation for a robust model of the environment under the distribution shift, which has been first studied by Representation Balancing MDP (RepBM). Although this prior work has shown promising results, there are a number of shortcomings that still hinder its applicability to practical tasks. In particular, we address the curse of horizon exhibited by RepBM, rejecting most of the pre-collected data in long-term tasks. We present a new objective for model learning motivated by recent advances in the estimation of stationary distribution corrections. This effectively overcomes the aforementioned limitation of RepBM, as well as naturally extending to continuous action spaces and stochastic policies. We also present an offline model-based policy optimization using this new objective, yielding the state-of-the-art performance in a representative set of benchmark offline RL tasks.

## 1 INTRODUCTION

Reinforcement learning (RL) has accomplished remarkable results in a wide range of domains, but its successes were mostly based on a large number of online interactions with the environment. However, in many real-world tasks, exploratory online interactions are either very expensive or dangerous (e.g. robotics, autonomous driving, and healthcare), and applying a standard online RL would be impractical. Consequently, the ability to optimize RL agents reliably without online interactions has been considered as a key to practical deployment, which is the main goal of batch RL, also known as offline RL (Fujimoto et al., 2019; Levine et al., 2020).

In an offline RL algorithm, accurate policy evaluation and reliable policy improvement are both crucial for the successful training of the agent. Evaluating policies in offline RL is essentially an off-policy evaluation (OPE) task, which aims to evaluate the target policy given the dataset collected from the behavior policy. The difference between the target and the behavior policies causes a *distribution shift* in the estimation, which needs to be adequately addressed for accurate policy evaluation. OPE itself is one of the long-standing hard problems in RL (Sutton et al., 1998; 2009; Thomas & Brunskill, 2016; Hallak & Mannor, 2017).

However, recent offline RL studies mainly focus on how to improve the policy conservatively while using a common policy evaluation technique without much considerations for the distribution shift, e.g. mean squared temporal difference error minimization or maximum-likelihood training of environment model (Fujimoto et al., 2019; Kumar et al., 2019; Yu et al., 2020). While conservative policy improvement helps the policy evaluation by reducing the off-policyness, we hypothesize that addressing the distribution shift explicitly during the policy evaluation can further improve the overall performance, since it can provide a better foundation for policy improvement.

To this end, we aim to explicitly address the distribution shift of the OPE estimator used in the offline RL algorithm. In particular, we focus on the model-based approach, where we train an

environment model robust to the distribution shift. One of the notable prior works is Representation Balancing MDP (RepBM) (Liu et al., 2018b), which regularizes the representation learning of the model to be invariant between the distributions. However, despite the promising results by RepBM, its *step-wise* estimation of the distance between the distributions has a few drawbacks that limit the algorithm from being practical: not only it assumes a discrete-action task where the target policy is deterministic, but it also performs poorly in long-term tasks due to the *curse of horizon* of step-wise importance sampling (IS) estimators (Liu et al., 2018a).

To address these limitations, we present the Representation Balancing with Stationary Distribution Estimation **(RepB-SDE)** framework, where we aim to learn a balanced representation by regularizing, in the representation space, the distance between the data distribution and the *discounted stationary distribution* induced by the target policy. Motivated by the recent advances in estimating stationary distribution corrections, we present a new representation balancing objective to train a model of the environment that no longer suffers from the *curse of horizon*. We empirically show that the model trained by the RepB-SDE objective is robust to the distribution shift for the OPE task, particularly when the difference between the target and the behavior is large. We also introduce a model-based offline RL algorithm based on the RepB-SDE framework and report its performance on the D4RL benchmark (Fu et al., 2020), showing the state-of-the-art performance in a representative set of tasks.

## 2 RELATED WORK

**Learning balanced representation**  Learning a representation invariant to specific aspects of data is an established method for overcoming distribution shift that arises in unsupervised domain adaptation (Ben-David et al., 2007; Zemel et al., 2013) and in causal inference from observational data (Shalit et al., 2017; Johansson et al., 2018). They have shown that imposing a bound on the generalization error under the distribution shift leads to the objective that learns a *balanced* representation such that the training and the test distributions look similar. RepBM (Liu et al., 2018b) can be seen as a direct extension to the sequential case, which encourages the representation to be invariant under the target and behavior policies in each timestep.

**Stationary distribution correction estimation (DICE)**  Step-wise importance sampling (IS) estimators (Precup, 2000) compute importance weights by taking the product of per-step distribution ratios. Consequently, these methods suffer from exponentially high variance in the lengths of trajectories, which is a phenomenon called the *curse of horizon* (Liu et al., 2018a). Recently, techniques of computing a stationary DIstribution Correction Estimation (DICE) have made remarkable progress that effectively addresses the curse of horizon (Liu et al., 2018a; Nachum et al., 2019a; Tang et al., 2020; Zhang et al., 2020; Mousavi et al., 2020). DICE has been also used to explicitly address the distribution shift in online model-free RL, by directly applying IS on the policy and action-value objectives (Liu et al., 2019; Gelada & Bellemare, 2019). We adopt one of the estimation techniques, DualDICE (Nachum et al., 2019a), to measure the distance between the stationary distribution and the data distribution in the representation space.

**Offline reinforcement learning**  There are extensive studies on improving standard online model-free RL algorithms (Mnih et al., 2015; Lillicrap et al., 2016; Haarnoja et al., 2018) for stable learning in the offline setting. The main idea behind them is to conservatively improve policy by (1) quantifying the uncertainty of value function estimate, e.g. using bootstrapped ensembles (Kumar et al., 2019; Agarwal et al., 2020), or/and (2) constraining the optimized target policy to be close to the behavior policy (i.e. behavior regularization approaches) (Fujimoto et al., 2019; Kumar et al., 2019; Wu et al., 2019; Lee et al., 2020). A notable exception is AlgaeDICE (Nachum et al., 2019b), which implicitly uses DICE to regularize the discounted stationary distribution induced by the target policy to be kept inside of the data support, similar to this work.

On the other hand, Yu et al. (2020) argued that the model-based approach can be advantageous due to its ability to generalize predictions on the states outside of the data support. They introduce MOPO (Yu et al., 2020), which uses truncated rollouts and penalized rewards for conservative policy improvement. MOReL (Kidambi et al., 2020) trains a state-action novelty detector and use it to penalize rewards in the data-sparse region. Matsushima et al. (2020), MOOSE (Swazinna et al., 2020)

and MBOP (Argenson & Dulac-Arnold, 2020) guide their policy optimization using the behavior policy, similar to the behavior regularization approaches.

Note that these aforementioned offline RL methods build on the standard approximate dynamic programming algorithm for action-value estimation (model-free) or on a maximum-likelihood environment model (model-based), without explicitly addressing the distribution shift in the estimator. In contrast, we augment the objective for model learning to obtain a robust model under the distribution shift, which is the first attempt for offline RL to the best of our knowledge.

## 3 PRELIMINARIES

A Markov Decision Process (MDP) is specified by a tuple $M = \langle \mathcal{S}, \mathcal{A}, T, R, d_0, \gamma \rangle$, consisting of state space $\mathcal{S}$, action space $\mathcal{A}$, transition function $T : \mathcal{S} \times \mathcal{A} \to \Delta(\mathcal{S})$, reward function $R : \mathcal{S} \times \mathcal{A} \to \Delta([0, r_{\max}])$, initial state distribution $d_0$, and discount rate $\gamma$. In this paper, we mainly focus on continuous state space $\mathcal{S} \subseteq \mathbb{R}^{d_s}$ and conduct experiments on both discrete action spaces $\mathcal{A} = \{a_0, ...a_{n_a}\}$ and continuous action spaces $\mathcal{A} \subseteq \mathbb{R}^{d_a}$. Given MDP $M$ and policy $\pi$, which is a (stochastic) mapping from state to action, the trajectory can be generated in the form of $s_0, a_0, r_0, s_1, a_1, r_1, ...$, where $s_0 \sim d_0$ and for each timestep $t \geq 0$, $a_t \sim \pi(s_t)$, $r_t \sim R(s_t, a_t)$, and $s_{t+1} \sim T(s_t, a_t)$. The goal of RL is to optimize or evaluate a policy, based on the *normalized* expected discounted return: $R^\pi \triangleq (1 - \gamma)\mathbb{E}_{M,\pi}\left[\sum_{t=0}^{\infty} \gamma^t r_t\right]$.

A useful and important concept throughout the paper is the *discounted stationary distribution*, which represents the long-term occupancy of states:

$$d^\pi(s, a) \triangleq (1 - \gamma) \sum_{t=0}^{\infty} \gamma^t \Pr(s_t = s, a_t = a | M, \pi).$$

From the definition, it can be observed that $R^\pi$ can be obtained by $R^\pi = \mathbb{E}_{(s,a) \sim d^\pi}[r(s, a)]$.

**Offline RL and off-policy evaluation**  In this paper, we focus on the *offline RL* problem where the agent can only access a static dataset $\mathcal{D} = \{(s_i, a_i, r_i, s_i')\}_{i=1}^{N}$ for the maximization of $R^\pi$. We consider a behavior-agnostic setting where we do not have any knowledge of the data collection process. We denote the empirical distribution of the dataset by $d^\mathcal{D}$.

Before improving policy, we first aim to better evaluate $R^\pi$ given a *target policy* $\pi$ and a static dataset $\mathcal{D}$, which corresponds to an off-policy evaluation (OPE) problem. We mainly focus on a model-based approach where the algorithm first estimates the unknown dynamics $\widehat{T}, \widehat{R}$ using the dataset $\mathcal{D}$. This defines an approximate MDP $\widehat{M} = \langle \mathcal{S}, \mathcal{A}, \widehat{T}, \widehat{R}, d_0, \gamma \rangle$, with the approximate expected discounted return $\widehat{R}^\pi \triangleq (1 - \gamma)\mathbb{E}_{\widehat{M},\pi}\left[\sum_{t=0}^{\infty} \gamma^t r_t\right]$ obtained from $\widehat{M}$. In this paper, we are interested in the MDP estimate $\widehat{M}$ that can effectively reduce the error in the evaluation of policy $\pi$, $|R^\pi - \widehat{R}^\pi|$.

In order to do so, we need to learn a good representation of a model that results in a small OPE error. We assume a bijective representation function $\phi : \mathcal{S} \times \mathcal{A} \to \mathcal{Z}$ where $\mathcal{Z} \subseteq \mathbb{R}^{d_z}$ is the representation space. We define the transition and the reward models in terms of the representation function $\phi$, i.e. $\widehat{T} = \widehat{T}_z \circ \phi$ and $\widehat{R} = \widehat{R}_z \circ \phi$. In practice, where we use a neural network for $\widehat{T}$ and $\widehat{R}$, $z$ can be chosen to be the output of an intermediate hidden layer, making $\phi$ represented by lower layers and $\widehat{T}_z, \widehat{R}_z$ by the remaining upper layers. We define $d_\phi^\pi(z)$ the discounted stationary distribution on $\mathcal{Z}$ induced by $d^\pi(s, a)$ under the representation function $z = \phi(s, a)$, and similarly for $d_\phi^\mathcal{D}(z)$.

## 4 REPRESENTATION BALANCING OFFLINE MODEL-BASED RL

### 4.1 GENERALIZATION ERROR BOUND FOR MODEL-BASED OFF-POLICY EVALUATION

We aim to construct a model $\widehat{M}$ from the dataset $\mathcal{D}$, which can accurately evaluate the policy $\pi$, by minimizing a good upper bound of policy evaluation error $|R^\pi - \widehat{R}^\pi|$. We define the following point-wise model loss for notational convenience:

$$\mathcal{E}_{\phi, \widehat{R}_z, \widehat{T}_z}(s, a) = c_R D_{TV}\left(R(r|s, a) \middle\| \widehat{R}_z(r|\phi(s, a))\right) + c_T D_{TV}\left(T(s'|s, a) \middle\| \widehat{T}_z(s'|\phi(s, a))\right),$$

where $c_R = 2(1 - \gamma)$ and $c_T = 2\gamma r_{\max}$. Then, we start by restating the *simulation lemma* (Kearns & Singh, 2002) to bound the policy evaluation error in terms of the point-wise model loss. The proof is available in Appendix A.

**Lemma 4.1.** *Given an MDP $M$ and its estimate $\widehat{M}$ with a bijective representation function $\phi$, i.e. $(\widehat{T}, \widehat{R}) = \langle \widehat{T}_z \circ \phi, \widehat{R}_z \circ \phi \rangle$, the policy evaluation error of a policy $\pi$ can be bounded by:*

$$\left| R^\pi - \widehat{R}^\pi \right| \leq \mathbb{E}_{(s,a)\sim d^\pi}\left[ \mathcal{E}_{\phi, \widehat{R}_z, \widehat{T}_z}(s, a) \right] \tag{1}$$

The Lemma 4.1 has a natural interpretation: if the model error is small in the states frequently visited by following the policy $\pi$, the resultant policy evaluation error will also be small. However, minimizing the RHS of Eq. (1) in the off-policy evaluation (OPE) task is generally intractable since the distribution $d^\pi$ is not directly accessible. Therefore, the common practice has been to construct a maximum-likelihood MDP using $\mathcal{D}$ while ignoring the distribution shift, but its OPE performance is not guaranteed.

Instead, we will derive a tractable upper bound on the policy evaluation error by eliminating the direct dependence on $d^\pi$ in Eq. (1). To this end, we adopt the distance metric between two distributions over representations $d_\phi^\pi$ and $d_\phi^\mathcal{D}$ that can bound their difference in expectations, which is the Integral Probability Metric (IPM) (Müller, 1997):

$$\text{IPM}_\mathcal{G}(p, q) = \sup_{g \in \mathcal{G}} \left| \mathbb{E}_{z \sim p}[g(z)] - \mathbb{E}_{z \sim q}[g(z)] \right|. \tag{2}$$

where particular choices of $\mathcal{G}$ make the IPM equivalent to different well-known distances of distributions, e.g. total variation distance or Wasserstein distance (Sriperumbudur et al., 2009).

**Theorem 4.2.** *Given an MDP $M$ and its estimate $\widehat{M}$ with a bijective representation function $\phi$, i.e. $(\widehat{T}, \widehat{R}) = \langle \widehat{T}_z \circ \phi, \widehat{R}_z \circ \phi \rangle$, assume that there exists a constant $B_\phi > 0$ and a function class $\mathcal{G} \triangleq \{g : \mathcal{Z} \to \mathbb{R}\}$ such that $\frac{1}{B_\phi} \mathcal{E}_{\phi, \widehat{R}_z, \widehat{T}_z}\left(\phi^{-1}(\cdot)\right) \in \mathcal{G}$. Then, for any policy $\pi$,*

$$\left| R^\pi - \widehat{R}^\pi \right| \leq \mathbb{E}_{(s,a)\sim d^\mathcal{D}}\left[ \mathcal{E}_{\phi, \widehat{R}_z, \widehat{T}_z}(s, a) \right] + B_\phi \text{IPM}_\mathcal{G}(d_\phi^\pi, d_\phi^\mathcal{D}) \tag{3}$$

This theorem is an adaptation of Lemma 1 of Shalit et al. (2017) to an *infinite horizon* model-based policy evaluation and can be derived by the definition of $\text{IPM}_\mathcal{G}(d_\phi^\pi, d_\phi^\mathcal{D})$ since it serves as an upper bound of the difference in the expectations of any function in $\mathcal{G}$. The first term in Eq. (3) corresponds to the fitness to the data following $d^\mathcal{D}$, while the second term serves as a regularizer. To see this, minimizing the second term would yield a near-constant representation function, which would be bad for the first term since it cannot distinguish states and actions well enough. It shows a natural trade-off between optimizing the model that fits data better and learning the representation that is invariant with respect to $d_\phi^\pi$ and $d_\phi^\mathcal{D}$.

Nevertheless, RHS of Eq. (3) still cannot be evaluated naively due to its dependence on $d^\pi$ in estimating the IPM. We address this challenge via a change of variable, which is known as a DualDICE trick (Nachum et al., 2019a). Define $\nu : \mathcal{Z} \to \mathbb{R}$ as an arbitrary function of state-action pairs that satisfies:

$$\nu\big(\phi(s, a)\big) \triangleq g\big(\phi(s, a)\big) + \gamma \mathbb{E}_{\substack{s' \sim T(s,a) \\ a' \sim \pi(s')}}\left[ \nu\big(\phi(s', a')\big) \right], \quad \forall (s, a) \in \mathcal{S} \times \mathcal{A}.$$

Then we can rewrite the IPM as:

$$\text{IPM}_\mathcal{G}(d_\phi^\pi, d_\phi^\mathcal{D}) = \sup_{g \in \mathcal{G}} \left| \mathbb{E}_{(s,a)\sim d^\pi}\left[ g\big(\phi(s, a)\big) \right] - \mathbb{E}_{(s,a)\sim d^\mathcal{D}}\left[ g\big(\phi(s, a)\big) \right] \right|$$

$$= \sup_{\nu \in \mathcal{F}} \left| (1 - \gamma) \mathbb{E}_{\substack{s \sim d_0 \\ a \sim \pi(s)}}\left[ \nu\big(\phi(s, a)\big) \right] - \mathbb{E}_{\substack{(s,a,s') \sim d^\mathcal{D} \\ a' \sim \pi(s')}}\left[ \nu\big(\phi(s, a)\big) - \gamma \nu\big(\phi(s', a')\big) \right] \right|, \tag{4}$$

$$\text{where } \mathcal{F} = \left\{ \nu : \nu(z) = \mathbb{E}_{T, \pi}\left[ \sum_{t=0}^\infty \gamma^t g(\phi(s_t, a_t)) \,\bigg|\, (s_0, a_0) = \phi^{-1}(z) \right], g \in \mathcal{G} \right\}.$$

In other words, we are now taking a supremum over the new function class $\mathcal{F}$, which captures a function that returns the expected discounted sum of $g(\phi(s_t, a_t))$ following the policy $\pi$ in an MDP

$M$ given an initial representation $z$. While it is now difficult to choose $\mathcal{F}$ from $\mathcal{G}$, Eq. (3) still can be kept valid by using a sufficiently rich function class for $\mathcal{F}$.

In this work, we choose $\mathcal{F}$ to be the family of functions in the unit ball in a reproducing kernel Hilbert space (RKHS) $\mathcal{H}_k$ with the kernel $k$, which allows the following closed-form formula (see Lemma A.3 in Appendix for details):

$$\mathrm{IPM}_{\mathcal{G}}(d_\phi^\pi, d_\phi^{\mathcal{D}})^2 = \mathbb{E}_{\substack{s_0 \sim d_0, a_0 \sim \pi(s_0), (s,a,s') \sim d^{\mathcal{D}}, a' \sim \pi(s') \\ \bar{s}_0 \sim d_0, \bar{a}_0 \sim \pi(\bar{s}_0), (\bar{s}, \bar{a}, \bar{s}') \sim d^{\mathcal{D}}, \bar{a}' \sim \pi(\bar{s}')}} \Bigg[ \tag{5}$$
$$k\big(\phi(s,a), \phi(\bar{s}, \bar{a})\big) + (1-\gamma)^2 k\big(\phi(s_0, a_0), \phi(\bar{s}_0, \bar{a}_0)\big) + \gamma^2 k\big(\phi(s', a'), \phi(\bar{s}', \bar{a}')\big)$$
$$- 2(1-\gamma) k\big(\phi(s_0, a_0), \phi(\bar{s}, \bar{a})\big) - 2\gamma k\big(\phi(s', a'), \phi(\bar{s}, \bar{a})\big) + 2\gamma(1-\gamma) k\big(\phi(s', a'), \phi(\bar{s}_0, \bar{a}_0)\big) \Bigg].$$

This completes our derivation for the tractable upper bound of policy evaluation error (Eq. (3)), whose direct dependence on $d^\pi$ is eliminated by Eq. (5). Finally, we can train a model by minimizing the upper bound that encourages us to learn balanced representation while improving data fitness, where the trained model can readily provide a model-based OPE.

The $\mathrm{IPM}_{\mathcal{G}}(d_\phi^\pi, d_\phi^{\mathcal{D}})^2$ in Eq. (5) can be estimated via finite random samples, and we denote its sampled-based estimator as $\widehat{\mathrm{IPM}}(d_\phi^\pi, d_\phi^{\mathcal{D}})^2$. We show in the following that, a valid upper bound can be established based on the sample-based estimators instead of the exact terms in the RHS of Eq. (3) under certain conditions.

**Theorem 4.3.** *Given an MDP $M$, its estimate $\widehat{M}$ with a bijective representation function $\phi$, i.e. $(\widehat{T}, \widehat{R}) = \langle \widehat{T}_z \circ \phi, \widehat{R}_z \circ \phi \rangle$, and an RKHS $\mathcal{H}_k \subset (\mathcal{Z} \to \mathbb{R})$ induced by a universal kernel $k$ such that $\sup_{z \in \mathcal{Z}} k(z, z) = \bar{k}$, assume that $f_{\phi, \widehat{R}_z, \widehat{T}_z}(z) = \mathbb{E}_{T, \pi} \Big[ \sum_{t=0}^\infty \gamma^t \mathcal{E}_{\phi, \widehat{R}_z, \widehat{T}_z}(s_t, a_t) \Big| (s_0, a_0) = \phi^{-1}(z) \Big] \in \mathcal{H}_k$ with $B_\phi = \| f_{\phi, \widehat{R}_z, \widehat{T}_z} \|_{\mathcal{H}_k}$ and the loss is bounded by $\bar{\mathcal{E}} = \sup_{s \in \mathcal{S}, a \in \mathcal{A}} \mathcal{E}_{\phi, \widehat{R}_z, \widehat{T}_z}(s, a)$. Let $n$ be the number of data in $\mathcal{D}$. With probability $1 - 2\delta$,*

$$\left| R^\pi - \widehat{R}^\pi \right| \leq \frac{1}{n} \sum_{(s,a) \in \mathcal{D}} \mathcal{E}_{\phi, \widehat{R}_z, \widehat{T}_z}(s, a) + B_\phi \widehat{\mathrm{IPM}}(d_\phi^\pi, d_\phi^{\mathcal{D}}) + \sqrt{\frac{\bar{\mathcal{E}}^2}{2n} \log \frac{1}{\delta}} + B_\phi \sqrt{\frac{\bar{k}}{n}} \left( 4 + \sqrt{8 \log \frac{3}{\delta}} \right).$$

This result can be proved by adapting the convergence results of the empirical estimate of the MMD (Gretton et al., 2012) and Hoeffding's inequality (Hoeffding, 1963). With the choice of an RKHS $\mathcal{H}_k$, we can now interpret $B_\phi$ as the RKHS norm $\| f_{\phi, \widehat{R}_z, \widehat{T}_z} \|_{\mathcal{H}_k}$, which captures the magnitude and the smoothness of the expected cumulative model loss $f_{\phi, \widehat{R}_z, \widehat{T}_z}$. In general, assuming smooth underlying dynamics, we can expect $B_\phi$ to be small when the model error is small. Furthermore, although $\bar{k}$ depends on the kernel function we use, we can always let $\bar{k} = 1$ and subsume it into $B_\phi$ as long as it is bounded, i.e. using $\tilde{B}_\phi \triangleq B_\phi \sqrt{\bar{k}}$. In the next section, we develop algorithms based on practical approximations of Eq. (3).

**Detailed comparison to RepBM** As previously stated, RepBM (Liu et al., 2018b) is a model-based finite-horizon OPE algorithm that trains the model to have balanced representation $\phi$, which is encouraged to be invariant under the target and behavior policies. Specifically, given the deterministic target policy $\pi$ and the behavior policy $\mu$, at each timestep $t$, it defines the *factual* distribution on $\mathcal{Z}$ given that the actions until timestep $t$ have been executed according to the policy $\pi$: $p_{\phi,t}^F(z) = \Pr(z_t | M, \mu, a_{0:t} = \pi(s_{0:t}))$ and the *counterfactual* distribution on $\mathcal{Z}$ given the same condition except the action at timestep $t$: $p_{\phi,t}^{CF}(z) = \Pr(z_t | M, \mu, a_{0:t-1} = \pi(s_{0:t-1}), a_t \neq \pi(s_t))$. Then, RepBM bounds the OPE error as,[1]

$$\left| R^\pi - \widehat{R}^\pi \right| \leq (1-\gamma) \sum_{t=0}^\infty \gamma^t \left( \mathbb{E}_{p(s_t | M, \mu, a_{0:t} = \pi(s_{0:t}))} [\mathcal{E}_{\phi, \widehat{R}_z, \widehat{T}_z}(s_t, \pi(s_t))] + B_{\phi,t} \mathrm{IPM}_{\mathcal{G}_t}(p_{\phi,t}^F, p_{\phi,t}^{CF}) \right).$$

Although RepBM achieves performance improvement over other OPE algorithms, we found a number of practical challenges: from the definition of the $\mathrm{IPM}_{\mathcal{G}_t}(p_{\phi,t}^F, p_{\phi,t}^{CF})$, it requires a discrete-action

---

[1] We adapted their formulation to the infinite horizon discounted MDP setting.

environment and a deterministic policy $\pi$, which cannot be met by many practical RL settings. In addition, since the sample-based estimation of $\mathrm{IPM}_{\mathcal{G}_t}(p_{\phi,t}^F, p_{\phi,t}^{CF})$ requires samples consistent with the policy $\pi$, i.e. $a_{0:t-1} = \pi(s_{0:t-1})$, the algorithm would reject exponentially many samples with respect to $t$, which is the *curse of horizon* (Liu et al., 2018a). When there is a large difference between the behavior and the target policies in long-term tasks, their implementation becomes close to using the maximum likelihood objective, which can also be observed empirically in our experiments. In contrast, our work is free from the abovementioned practical limitations by performing balancing between the discounted stationary distribution $d^\pi$ and the data distribution $d^{\mathcal{D}}$, leveraging the recent advances in stationary distribution correction estimation (i.e. the DualDICE trick) to overcome the difficulties pertinent to the expectation concerning $d^\pi$ required to evaluate the IPM in the objective.

## 4.2 REPRESENTATION BALANCING WITH STATIONARY DISTRIBUTION ESTIMATION

In the following, we describe algorithms for OPE and offline RL based on the practical approximations to Eq. (3), which we call the RepB-SDE framework.

**Objective for off-policy evaluation**    As we mentioned earlier, we aim to minimize the upper bound of OPE error $|R^\pi - \widehat{R}^\pi|$ specified in Theorem 4.2. To make the RHS of Eq. (3) tractable for optimization, we replace the intractable total variation distance with KL-divergence, which can be easily minimized by maximizing the data log-likelihood. We also replace the IPM with its sample-based estimator to obtain the learning objective:

$$\min_{\phi, \widehat{R}_z, \widehat{T}_z} \frac{1}{n} \sum_{(s,a,s',r) \in \mathcal{D}} \Big[ \underbrace{-\log \widehat{R}_z\big(r|\phi(s,a)\big)}_{D_{\mathrm{KL}}(R(r|s,a)||\widehat{R}_z(r|\phi(s,a)))} \quad \underbrace{-\log \widehat{T}_z\big(s'|\phi(s,a)\big)}_{D_{\mathrm{KL}}(T(s'|s,a)||\widehat{T}_z(s'|\phi(s,a)))} \Big] + \alpha_M \widehat{\mathrm{IPM}}(d_\phi^\pi, d_\phi^{\mathcal{D}}) \quad (6)$$

The constant $B_\phi$ in Theorem 4.2 depends on the function classes and cannot be estimated, and thus, we replace it with a tunable hyperparameter $\alpha_M$ that balances between data fitness and representation invariance. Remark that $\alpha_M = 0$ recovers the simple maximum-likelihood objective.

By simulating the target policy $\pi$ under the environment model $(\widehat{T}, \widehat{R})$ obtained by minimizing Eq. (6), it is possible to perform a model-based OPE that approximately minimizes the upper bound of OPE error.

**Objectives for offline model-based RL**    By rearranging Eq. (3), we have,

$$R^\pi \geq \widehat{R}^\pi - \mathbb{E}_{(s,a) \sim \mathcal{D}} \Big[ \mathcal{E}_{\phi, \widehat{R}_z, \widehat{T}_z}(s,a) \Big] - B_\phi \mathrm{IPM}_{\mathcal{G}}(d_\phi^\pi, d_\phi^{\mathcal{D}}). \quad (7)$$

Then, we can maximize the RHS of Eq. (7) to get the model and the policy that maximizes the lower bound of true return $R^\pi$. Similar to the derivation of Eq. (6), we replace the total variation distance by KL-divergence to obtain the following learning objectives:

$$\mathcal{L}_{\widehat{M}}(\widehat{M}, \pi, \alpha_M) = \mathbb{E}_{d^{\mathcal{D}}} \Big[ -\log \widehat{R}_z\big(r|\phi(s,a)\big) - \log \widehat{T}_z\big(s'|\phi(s,a)\big) \Big] + \alpha_M \mathrm{IPM}_{\mathcal{G}}(d_\phi^\pi, d_\phi^{\mathcal{D}}), \quad (8)$$

$$\mathcal{J}_\pi(\pi, \widehat{M}, \alpha_\pi) = \mathbb{E}_{\widehat{M}, \pi} \Big[ \sum_{t=0}^\infty \gamma^t r_t \Big] - \alpha_\pi \mathrm{IPM}_{\mathcal{G}}(d_\phi^\pi, d_\phi^{\mathcal{D}}). \quad (9)$$

where the expectation in Eq. (9) can be optimized using various model-based RL algorithms, e.g. with planning (Chua et al., 2018) or using a model-free learner (Janner et al., 2019). By iterating between the minimization of $\mathcal{L}_{\widehat{M}}$ with respect to $\widehat{M}$ and the maximization of $\mathcal{J}_\pi$ with respect to $\pi$ by stochastic gradient method, it is possible to perform offline model-based RL that approximately maximizes the lower bound of the true return $R^\pi$.

**Implementation details**    Following the recent practice (Chua et al., 2018; Janner et al., 2019; Yu et al., 2020), we model the dynamics $(\widehat{T}, \widehat{R})$ using a bootstrap ensemble of neural networks. To optimize a policy based on the objective, we perform full rollouts (until reaching the terminal states or maximum timesteps) using learned dynamics $(\widehat{T}, \widehat{R})$. During obtaining the rollouts, we pessimistically augment the estimated reward function using the penalty proportional to the bootstrapped uncertainty, which helped the algorithm to perform robustly. We suspect the difficulty in calculating

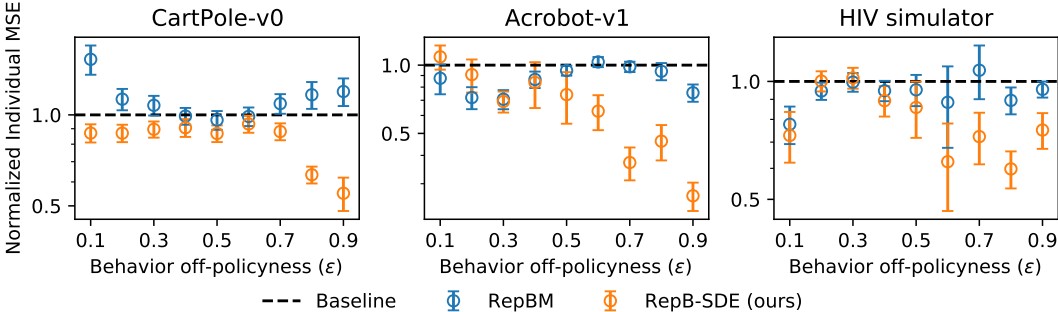

Figure 1: The OPE results of different model learning algorithms with varying off-policyness on the x-axis. The y-axis plots the normalized individual MSE on test trajectories where the performance of the baseline model is set to 1. The tasks used are infinite-horizon discounted environments ($\gamma = 0.98$ (HIV), $\gamma = 0.99$ (others)), where we truncated at $t = 1000$. The experiments are repeated 200 times, and the error bars indicate 95% confidence intervals.

accurate IPM estimation is what makes the additional pessimism beneficial. We store the generated experiences to a separate dataset $\widehat{\mathcal{D}}$ and update the policy $\pi$ with IPM-regularized soft actor-critic (SAC) (Haarnoja et al., 2018) using samples from both datasets $\mathcal{D} \cup \widehat{\mathcal{D}}$ similar to MBPO (Janner et al., 2019).

Since the presented model objective requires a policy $\pi$ to perform a balancing, we initially trained the model and the policy using $\alpha_M = \alpha_\pi = 0$: by $\widehat{M}_0 = \arg\min_{\widehat{M}} \mathcal{L}_{\widehat{M}}(\widehat{M}, \cdot, 0)$ and $\pi_0 = \arg\min_\pi \mathcal{L}_\pi(\pi, \widehat{M}_0, 0)$. Then, we retrained the model and the policy using $\pi_0$: $\widehat{M}_1 = \arg\min_{\widehat{M}} \mathcal{L}_{\widehat{M}}(\widehat{M}, \pi_0, \alpha_M)$ and $\pi_1 = \arg\min_\pi \mathcal{L}_\pi(\pi, \widehat{M}_1, \alpha_\pi)$ for some non-negative $\alpha_M$ and $\alpha_\pi$. While it is desirable to repeat the optimization of the model and the policy until convergence, we did not observe significant improvement after the first iteration and reported the performance of the policy after the first iteration, $\pi_1$.

## 5 EXPERIMENTS

We demonstrate the effectiveness of the RepB-SDE framework, by comparing the OPE performance of Eq. (6) to that of RepBM and evaluating the presented model-based offline RL algorithm on the benchmarks. The code used to produce the results is available online.[2] A detailed description of the experiments can be found in Appendix B.

### 5.1 MODEL-BASED OFF-POLICY EVALUATION

For the sake of comparison with RepBM, we test our OPE algorithm on three continuous-state *discrete*-action tasks where the goal is to evaluate a *deterministic* target policy. We trained a sub-optimal deterministic target policy $\pi$ and used an $\epsilon$-greedy policy with various values of $\epsilon$ as the data collection policy. In each experiment, we trained environment models for a fixed number of epochs concerning three different objectives: simple maximum-likelihood baseline, step-wise representation balancing objective used in RepBM (Liu et al., 2018b), and the presented OPE objective of RepB-SDE (Eq. (6)). We measured the individual mean squared error (Liu et al., 2018b).

The normalized error of each algorithm, relative to the error of the baseline valued at 1, is presented in Figure 1. We can observe that the presented objective of RepB-SDE can reduce the OPE error from the baseline significantly, outperforming RepBM in most of the cases. As the off-policyness between the policies ($\epsilon$) increases, representation balancing algorithms should more benefit compared to the maximum-likelihood baseline in principle. However, the result shows that the performance of RepBM merely increases due to the increased sample rejection rate under large $\epsilon$.

---

[2]https://github.com/dlqudwns/repb-sde

Table 1: Normalized scores on D4RL MuJoCo benchmark datasets (Fu et al., 2020) where the score of 0 corresponds to a random policy and 100 corresponds to a converged SAC policy. All results (except MF, which is taken from Fu et al. (2020) and Kumar et al. (2020)) are averaged over 5 runs, where $\pm$ denotes the standard error. The highest scores are highlighted with boldface.

| Dataset type | Environment | RepB-SDE (ours) | RP | Base | MOPO | MF | BC |
|---|---|---|---|---|---|---|---|
| Random | Walker2d | **21.1** $\pm$ 1.0 | 18.4 | 16.4 | 1.3 | 7.3 [BEAR] | 0.0 |
| Medium | Walker2d | 72.1 $\pm$ 1.9 | 56.3 | 5.5 | -0.1 | **81.1** [BRAC] | 7.7 |
| Med-Replay | Walker2d | **49.8** $\pm$ 11.4 | 41.1 | 6.2 | 47.8 | 26.7 [CQL] | 8.0 |
| Med-Expert | Walker2d | 88.8 $\pm$ 6.9 | 72.6 | 51.7 | 32.4 | **98.7** [CQL] | 3.3 |
| Random | Hopper | 8.6 $\pm$ 1.0 | 8.3 | 8.3 | 9.1 | **12.2** [BRAC] | 9.0 |
| Medium | Hopper | 34.0 $\pm$ 2.8 | 27.5 | 19.8 | 19.2 | **58.0** [CQL] | 34.5 |
| Med-Replay | Hopper | 62.2 $\pm$ 6.7 | 49.8 | 32.9 | **80.8** | 48.6 [CQL] | 13.2 |
| Med-Expert | Hopper | 82.6 $\pm$ 7.0 | 74.0 | 19.1 | 23.2 | **111.0** [CQL] | 41.9 |
| Random | HalfCheetah | 32.9 $\pm$ 1.1 | 31.3 | 26.1 | 29.9 | **35.4** [CQL] | -2.4 |
| Medium | HalfCheetah | **49.1** $\pm$ 0.3 | 47.3 | 22.1 | 45.0 | 46.3 [BRAC] | 31.0 |
| Med-Replay | HalfCheetah | **57.5** $\pm$ 0.8 | 53.6 | 54.0 | 53.8 | 47.7 [BRAC] | 14.7 |
| Med-Expert | HalfCheetah | 55.4 $\pm$ 8.3 | 36.7 | 31.9 | **91.0** | 64.7 [BCQ] | 29.5 |

## 5.2 OFFLINE REINFORCEMENT LEARNING WITH REPRESENTATION BALANCED MODEL

We evaluate the offline model-based RL algorithm presented in Section 4.2 on a subset of datasets in the D4RL benchmark (Fu et al., 2020): using four types of datasets (Random, Medium, Medium-Replay, and Medium-Expert) from three different MuJoCo environments (HalfCheetah-v2, Hopper-v2, and Walker2d-v2) (Todorov et al., 2012). **Random** dataset contains $10^6$ experience tuples from a random policy. **Medium** dataset contains $10^6$ experience tuples from a policy trained to approximately $1/3$ the performance of the expert, which is an agent trained to completion with SAC. **Med-Replay** dataset contains $10^5$ ($2 \times 10^5$ for Walker2d-v2) experience tuples, which are from the replay buffer of a policy trained up to the performance of the medium agent. **Med-Expert** dataset is a **Medium** dataset combined with $10^6$ samples from the expert. This experimental setting exactly follows that of (Yu et al., 2020; Argenson & Dulac-Arnold, 2020).

The normalized score of each algorithm is presented in Table 1. **MF** denotes the best score from offline model-free algorithms (taken from Fu et al. (2020) and Kumar et al. (2020)), including SAC (Haarnoja et al., 2018), BCQ (Fujimoto et al., 2019), BEAR (Kumar et al., 2019), BRAC (Wu et al., 2019), AWR (Peng et al., 2019), cREM (Agarwal et al., 2020), AlgaeDICE (Nachum et al., 2019b), and CQL (Kumar et al., 2020). The actual algorithm that achieves the reported score is presented next to the numbers. **Base** shows the performance of the most naive baseline, which attempts to maximize the estimated policy return under the maximum-likelihood model. **RP** denotes the performance of **Base** equipped with the appropriate reward penalty using the bootstrapped uncertainty of the model, which is equivalent to $\pi_0$ described in Section 4.2. **RepB-SDE** denotes the performance after a single iteration of our algorithm, corresponding to $\pi_1$. We also provide **BC**, the performance of direct behavior cloning from the data, and **MOPO** (Yu et al., 2020), an offline model-based RL algorithm that optimizes policy based on truncated rollouts with the heuristic reward penalty.

The significant gap between **RepB-SDE** and **RP** in the results shows the advantage brought by our framework that encourages balanced representation. While our approach was less successful on some of the datasets (mostly on the Hopper-v2 environment), we hypothesize that the conservative training techniques: the behavior regularization approaches exploited in the model-free algorithms, the rollout truncation technique in MOPO, and the pessimistic training based on the bootstrapped uncertainty estimates adopted in our algorithm exhibit their strengths in different datasets. For example, it may be the case that the ensemble models are overconfident especially in Hopper-v2, and should be regularized with more explicit methods. Nevertheless, we emphasize that the presented framework can be used jointly with any other conservative training technique to improve their performance.

## 6 CONCLUSION AND FUTURE WORK

In this paper, we presented RepB-SDE, a framework for balancing the model representation with stationary distribution estimation, aiming at obtaining a model robust to the distribution shift that arises in off-policy and offline RL. We started from the theoretical observation that the model-based policy evaluation error can be upper-bounded by the data fitness and the distance between two distributions in the representation space. Motivated by the bound, we presented a model learning objective for off-policy evaluation and model-based offline policy optimization. RepB-SDE can be seen as an extension of RepBM, which addresses the *curse of horizon* by leveraging the recent advances in stationary distribution correction estimation (i.e. the DualDICE trick). Using stationary distribution also frees us from other limitations of RepBM to be applied to more practical settings. To the best of our knowledge, it is the first attempt to introduce an augmented objective for the learning of model robust to a specific distribution shift in offline RL.

In the experiments, we empirically demonstrated that we can significantly reduce the OPE error from the baseline, outperforming RepBM in most cases. We also showed that the robust model also helps in the offline model-based policy optimization, yielding the state-of-the-art performance in a representative set of D4RL benchmarks. We emphasize that our approach can be directly adopted in many other model-based offline RL algorithms.

There are a number of promising directions for future work. Most importantly, we have not leveraged the learned representation in the policy when optimizing the policy, yet it is very natural to do so. We can easily incorporate the representation into the policy by assuming energy-based models, but this would make the computation of entropy intractable in entropy-regularized policy optimization algorithms. It would be also interesting to see if the proposed framework for learning balanced representation can benefit off-policy (and offline) model-free methods.

### ACKNOWLEDGMENTS

This work was supported by the National Research Foundation (NRF) of Korea (NRF-2019M3F2A1072238 and NRF-2019R1A2C1087634), and the Ministry of Science and Information communication Technology (MSIT) of Korea (IITP No. 2019-0-00075, IITP No. 2020-0-00940 and IITP No. 2017-0-01779 XAI).

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

## A    PROOFS

**Lemma 4.1.** *Given an MDP $M$ and its estimate $\widehat{M}$ with a bijective representation function $\phi$, i.e.* $(\widehat{T}, \widehat{R}) = \langle \widehat{T}_z \circ \phi, \widehat{R}_z \circ \phi \rangle$, *the policy evaluation error of a policy $\pi$ can be bounded by:*

$$\left| R^\pi - \widehat{R}^\pi \right| \le \mathbb{E}_{(s,a) \sim d^\pi} \left[ \mathcal{E}_{\phi, \widehat{R}_z, \widehat{T}_z}(s,a) \right] \tag{1}$$

*Proof.* We define a value function $V^\pi(s) = \mathbb{E}_{M,\pi}[\sum_{t=0}^\infty \gamma^t r_t | s_0 = s]$, the expected discounted return starting from the state $s$, for the following proof. Due to its definition, we can write $R^\pi = (1-\gamma) \mathbb{E}_{s \sim d_0}[V^\pi(s)]$. The following recursive equation also holds:

$$V^\pi(s) = \mathbb{E}_{a \sim \pi(s)} \left[ r(s,a) + \gamma \mathbb{E}_{s' \sim T(s,a)}[V(s')] \right],$$

and similarly with an approximate value function $\widehat{V}^\pi(s) = \mathbb{E}_{\widehat{M},\pi}[\sum_{t=0}^\infty \gamma^t r_t | s_0 = s]$ given an MDP estimate $\widehat{M}$. Then,

$$\frac{1}{1-\gamma} \left| R^\pi - \widehat{R}^\pi \right| = \left| \mathbb{E}_{d_0} \left[ V^\pi(s_0) - \widehat{V}^\pi(s_0) \right] \right|$$

$$= \left| \mathbb{E}_{d_0,\pi}[r_0 - \widehat{r}_0] + \gamma \mathbb{E}_{d_0} \left[ \int \left( V^\pi(s_1) T(s_1|s_0,a_0) - \widehat{V}^\pi(s_1) \widehat{T}(s_1|s_0,a_0) \right) \pi(a_0|s_0) ds_1 da_0 \right] \right|$$

$$= \left| \mathbb{E}_{d_0,\pi}[r_0 - \widehat{r}_0] + \gamma \mathbb{E}_{d_0} \left[ \int \left( V^\pi(s_1) T(s_1|s_0,a_0) - \widehat{V}^\pi(s_1) T(s_1|s_0,a_0) \right. \right. \right.$$

$$\left. \left. \left. + \widehat{V}^\pi(s_1) T(s_1|s_0,a_0) - \widehat{V}^\pi(s_1) \widehat{T}(s_1|s_0,a_0) \right) \pi(a_0|s_0) ds_1 da_0 \right] \right|$$

$$= \left| \mathbb{E}_{d_0,\pi}[r_0 - \widehat{r}_0] + \gamma \mathbb{E}_{d_0} \left[ \int \underbrace{\left( V^\pi(s_1) - \widehat{V}^\pi(s_1) \right)}_{\text{This forms a recursive equation}} T(s_1|s_0,a_0) \pi(a_0|s_0) ds_1 da_0 \right] \right.$$

$$\left. + \gamma \mathbb{E}_{d_0} \left[ \int \widehat{V}^\pi(s_1) \left( T(s_1|s_0,a_0) - \widehat{T}(s_1|s_0,a_0) \right) \pi(a_0|s_0) ds_1 da_0 \right] \right|$$

$$= \left| \mathbb{E}_{(s,a) \sim d^\pi} \left[ \int \left( R(r|s,a) - \widehat{R}(r|s,a) \right) dr + \gamma \int \widehat{V}^\pi(s') \left( T(s'|s,a) - \widehat{T}(s'|s,a) \right) ds' \right] \right|$$

$$\le \mathbb{E}_{(s,a) \sim d^\pi} \left[ \int \left| R(r|s,a) - \widehat{R}(r|s,a) \right| dr + \gamma \int \widehat{V}^\pi(s') \left| T(s'|s,a) - \widehat{T}(s'|s,a) \right| ds' \right]$$

$$\le \mathbb{E}_{(s,a) \sim d^\pi} \left[ \int \left| R(r|s,a) - \widehat{R}(r|s,a) \right| dr + \gamma \int \frac{r_{\max}}{1-\gamma} \left| T(s'|s,a) - \widehat{T}(s'|s,a) \right| ds' \right]$$

$$= 2 \mathbb{E}_{(s,a) \sim d^\pi} \left[ D_{TV}(R(r|s,a) || \widehat{R}_z(r|\phi(s,a))) \right]$$

$$+ \frac{2\gamma r_{\max}}{1-\gamma} \mathbb{E}_{(s,a) \sim d^\pi} \left[ D_{TV}(T(s'|s,a) || \widehat{T}_z(s'|\phi(s,a))) \right]$$

$\square$

**Theorem 4.2.** *Given an MDP $M$ and its estimate $\widehat{M}$ with a bijective representation function $\phi$, i.e.* $(\widehat{T}, \widehat{R}) = \langle \widehat{T}_z \circ \phi, \widehat{R}_z \circ \phi \rangle$, *assume that there exists a constant $B_\phi > 0$ and a function class* $\mathcal{G} \triangleq \{g : \mathcal{Z} \to \mathbb{R}\}$ *such that $\frac{1}{B_\phi} \mathcal{E}_{\phi, \widehat{R}_z, \widehat{T}_z}(\phi^{-1}(\cdot)) \in \mathcal{G}$. Then, for any policy $\pi$,*

$$\left| R^\pi - \widehat{R}^\pi \right| \le \mathbb{E}_{(s,a) \sim d^{\mathcal{D}}} \left[ \mathcal{E}_{\phi, \widehat{R}_z, \widehat{T}_z}(s,a) \right] + B_\phi \text{IPM}_{\mathcal{G}}(d_\phi^\pi, d_\phi^{\mathcal{D}}) \tag{3}$$

*Proof.* From Lemma 4.1, we directly have:

$$\left| R^\pi - \widehat{R}^\pi \right| \le \mathbb{E}_{(s,a) \sim d^{\mathcal{D}}} \left[ \mathcal{E}_{\phi, \widehat{R}_z, \widehat{T}_z}(s,a) \right] + \int \mathcal{E}_{\phi, \widehat{R}_z, \widehat{T}_z}(s,a) \left( d^\pi(s,a) - d^{\mathcal{D}}(s,a) \right) ds da.$$

Then,

$$\int \mathcal{E}_{\phi,\widehat{R}_z,\widehat{T}_z}(s,a)\left(d^\pi(s,a) - d^{\mathcal{D}}(s,a)\right) dsda$$

$$= \int \mathcal{E}_{\phi,\widehat{R}_z,\widehat{T}_z}(\phi^{-1}(z))\left(d^\pi(z) - d^{\mathcal{D}}\left(\phi^{-1}(\phi^{-1}(z))\right)\right)\left|\frac{d(s,a)}{dz}\right| dz$$

$$= \int \mathcal{E}_{\phi,\widehat{R}_z,\widehat{T}_z}(\phi^{-1}(z))\left(d^\pi_\phi(z) - d^{\mathcal{D}}_\phi(z)\right) dz$$

$$\leq B_\phi \left|\int \frac{1}{B_\phi}\mathcal{E}_{\phi,\widehat{R}_z,\widehat{T}_z}(\phi^{-1}(z))\left(d^\pi_\phi(z) - d^{\mathcal{D}}_\phi(z)\right) dz\right| \qquad (B_\phi > 0)$$

$$\leq B_\phi \sup_{g\in\mathcal{G}}\left|\int g(z)\left(d^\pi_\phi(z) - d^{\mathcal{D}}_\phi(z)\right) dz\right| \qquad \left(\frac{1}{B_\phi}\mathcal{E}_{\phi,\widehat{R}_z,\widehat{T}_z}(\phi^{-1}(z)) \in \mathcal{G}\right)$$

$$= B_\phi \mathrm{IPM}_{\mathcal{G}}\left(d^\pi_\phi, d^{\mathcal{D}}_\phi\right)$$

$$\square$$

We state some previous results, which are required for further proof.

**Theorem A.1** (McDiarmid's inequality (McDiarmid, 1989)). *Let $\{X_i\}_{i=1}^n$ be independent random variables taking values in set $\mathcal{X}$, and assume that $f : \mathcal{X}^n \to \mathbb{R}$ satisfies*

$$\sup_{\{x_i\}_{i=1}^n \in \mathcal{X}^n, \tilde{x}\in\mathcal{X}} |f(\{x_i\}_{i=1}^n) - f(x_1,...,x_{i-1},\tilde{x},x_{i+1},...,x_n)| \leq c_i. \qquad (10)$$

*Then for every $\epsilon > 0$,*

$$\Pr\{f(\{X_i\}_{i=1}^n) - \mathbb{E}_{\{X_i\}_{i=1}^n}[f(\{X_i\}_{i=1}^n)] \geq \epsilon\} \leq \exp\left(-\frac{2\epsilon^2}{\sum_{i=1}^n c_i^2}\right). \qquad (11)$$

**Lemma A.2** (Rademacher complexity of RKHS (Bartlett & Mendelson, 2002)). *Let $\mathcal{F}$ be a unit ball in a universal RKHS on the compact domain $\mathcal{X}$, with kernel bounded according to $0 \leq k(x,x') \leq \bar{k}$. Let $\{x_i\}_{i=1}^n$ be an i.i.d. sample of size $n$ drawn according to a probability measure $p$ on $\mathcal{X}$, and let $\sigma_i$ be i.i.d. and take values in $\{-1,1\}$ with equal probability. The Rademacher complexity, which is defined as below, is upper bounded as:*

$$R_n(\mathcal{F}) \triangleq \mathbb{E}_{\{x_i\}_{i=1}^n,\sigma}\sup_{f\in\mathcal{F}}\left|\frac{1}{n}\sum_{i=1}^n \sigma_i f(x_i)\right| \leq \sqrt{\frac{\bar{k}}{n}}. \qquad (12)$$

*The upper bound is followed by Lemma 22 of (Bartlett & Mendelson, 2002).*

Now we prove the following using the results above.

**Lemma A.3.** *Let $\mathcal{H}_k$ be a RKHS associated with universal kernel $k(\cdot,\cdot)$. Let $\langle\cdot,\cdot\rangle_{\mathcal{H}_k}$ be the inner product of $\mathcal{H}_k$, which satisfies the reproducing property $\nu(z) = \langle\nu, k(\cdot,z)\rangle_{\mathcal{H}_k}$. When $\mathcal{G}$ is chosen such that*

$$\mathcal{G} = \left\{g \in (\mathcal{Z}\to\mathbb{R}) : g(z) = \nu(z) - \gamma\mathbb{E}_{\substack{s'\sim T(\phi^{-1}(z))\\a'\sim\pi(s')}}[\nu(\phi(s',a'))], \nu \in (\mathcal{Z}\to\mathbb{R}), \langle\nu,\nu\rangle_{\mathcal{H}_k} \leq 1\right\},$$

*the $\mathrm{IPM}_{\mathcal{G}}(d^\pi_\phi, d^{\mathcal{D}}_\phi)$ has the following closed form definition:*

$$\mathrm{IPM}_{\mathcal{G}}(d^\pi_\phi, d^{\mathcal{D}}_\phi)^2 = \mathbb{E}_{\substack{s_0\sim d_0,a_0\sim\pi(s_0),(s,a,s')\sim d^{\mathcal{D}},a'\sim\pi(s')\\ \bar{s}_0\sim d_0,\bar{a}_0\sim\pi(\bar{s}_0),(\bar{s},\bar{a},\bar{s}')\sim d^{\mathcal{D}},\bar{a}'\sim\pi(\bar{s}')}}\Bigg[$$

$$k(\phi(s,a),\phi(\bar{s},\bar{a})) + (1-\gamma)^2 k(\phi(s_0,a_0),\phi(\bar{s}_0,\bar{a}_0)) + \gamma^2 k(\phi(s',a'),\phi(\bar{s}',\bar{a}'))$$

$$- 2(1-\gamma)k(\phi(s_0,a_0),\phi(\bar{s},\bar{a})) - 2\gamma k(\phi(s',a'),\phi(\bar{s},\bar{a})) + 2\gamma(1-\gamma)k(\phi(s',a'),\phi(\bar{s}_0,\bar{a}_0))\Bigg].$$

*Furthermore, suppose that $\bar{k} \triangleq \sup_{z\in\mathcal{Z}} k(z,z)$. The estimator $\widehat{\mathrm{IPM}}(d^\pi_\phi, d^{\mathcal{D}}_\phi)^2$, which is the sample-based estimation of $\mathrm{IPM}_{\mathcal{G}}(d^\pi_\phi, d^{\mathcal{D}}_\phi)^2$ from $n$ samples, satisfies with probability at least $1 - \delta$,*

$$\left|\mathrm{IPM}_{\mathcal{G}}(d^\pi_\phi, d^{\mathcal{D}}_\phi) - \widehat{\mathrm{IPM}}(d^\pi_\phi, d^{\mathcal{D}}_\phi)\right| \leq \sqrt{\frac{\bar{k}}{n}}\left(4 + \sqrt{8\log\frac{3}{\delta}}\right).$$

*Proof.* In the below we write in shorthand, $\mathrm{IPM}_{\mathcal{G}}$ to denote $\mathrm{IPM}_{\mathcal{G}}(d_\phi^\pi, d_\phi^{\mathcal{D}})$. As in Eq. (4), we can rewrite the IPM term as:

$$\mathrm{IPM}_{\mathcal{G}} = \sup_{\nu \in \mathcal{F}} \left| (1-\gamma) \mathbb{E}_{\substack{s \sim d_0 \\ a \sim \pi(s)}} [\nu(\phi(s,a))] - \mathbb{E}_{\substack{(s,a,s') \sim d^{\mathcal{D}} \\ a' \sim \pi(s')}} [\nu(\phi(s,a)) - \gamma \nu(\phi(s',a'))] \right|,$$

and $\mathcal{F}$ here becomes $\mathcal{F} = \{\nu \in (\mathcal{Z} \to \mathbb{R}) : \langle \nu, \nu \rangle_{\mathcal{H}_k} \leq 1\}$, a unit ball in RKHS $\mathcal{H}_k$. Using the reproducing property of $\mathcal{H}_k$:

$$\mathrm{IPM}_{\mathcal{G}}^2 = \left\{ \sup_{\nu \in \mathcal{F}} \left| (1-\gamma) \mathbb{E}_{\substack{s \sim d_0 \\ a \sim \pi(s)}} [\nu(\phi(s,a))] - \mathbb{E}_{\substack{(s,a,s') \sim d^{\mathcal{D}} \\ a' \sim \pi(s')}} [\nu(\phi(s,a)) - \gamma \nu(\phi(s',a'))] \right| \right\}^2$$

$$= \sup_{\nu \in \mathcal{F}} \left\{ (1-\gamma) \mathbb{E}_{\substack{s \sim d_0 \\ a \sim \pi(s)}} [\nu(\phi(s,a))] - \mathbb{E}_{\substack{(s,a,s') \sim d^{\mathcal{D}} \\ a' \sim \pi(s')}} [\nu(\phi(s,a)) - \gamma \nu(\phi(s',a'))] \right\}^2$$

$$= \sup_{\nu \in \mathcal{F}} \left\{ (1-\gamma) \mathbb{E}_{\substack{s \sim d_0 \\ a \sim \pi(s)}} [\langle \nu, k(\cdot, \phi(s,a)) \rangle_{\mathcal{H}_k}] \right.$$

$$\left. - \mathbb{E}_{\substack{(s,a,s') \sim d^{\mathcal{D}} \\ a' \sim \pi(s')}} [\langle \nu, k(\cdot, \phi(s,a)) \rangle_{\mathcal{H}_k} - \gamma \langle \nu, k(\cdot, \phi(s',a')) \rangle_{\mathcal{H}_k}] \right\}^2$$

$$= \sup_{\nu \in \mathcal{F}} \langle \nu, \nu^* \rangle_{\mathcal{H}_k}^2,$$

where

$$\nu^*(\cdot) = (1-\gamma) \mathbb{E}_{\substack{s \sim d_0 \\ a \sim \pi(s)}} [k(\cdot, \phi(s,a))] - \mathbb{E}_{\substack{(s,a,s') \sim d^{\mathcal{D}} \\ a' \sim \pi(s')}} [k(\cdot, \phi(s,a)) - \gamma k(\cdot, \phi(s',a'))].$$

Due to the Cauchy-Schwarz inequality and $\langle \nu, \nu \rangle_{\mathcal{H}_k} \leq 1, \ \forall \nu \in \mathcal{F}$,

$$\langle \nu, \nu^* \rangle_{\mathcal{H}_k}^2 \leq \langle \nu, \nu \rangle_{\mathcal{H}_k} \langle \nu^*, \nu^* \rangle_{\mathcal{H}_k} \leq \langle \nu^*, \nu^* \rangle_{\mathcal{H}_k}$$

holds and $\sup_{\nu \in \mathcal{F}} \langle \nu, \nu^* \rangle_{\mathcal{H}_k}^2 = \langle \nu^*, \nu^* \rangle_{\mathcal{H}_k}$. Using the property that $\langle k(\cdot, z), k(\cdot, \bar{z}) \rangle_{\mathcal{H}_k} = k(z, \bar{z})$, we can derive the closed form expression in the lemma from $\langle \nu^*, \nu^* \rangle_{\mathcal{H}_k}$.

Now we prove the error bound of the estimator. First, we divide $\mathrm{IPM}_{\mathcal{G}}(d_\phi^\pi, d_\phi^{\mathcal{D}})$ into three parts:

$$\mathrm{IPM}_{\mathcal{G}}(d_\phi^\pi, d_\phi^{\mathcal{D}}) = \sup_{\nu \in \mathcal{F}} |f_1(\nu) + f_2(\nu) + f_3(\nu)|$$

$$\text{where} \quad f_1(\nu) = (1-\gamma) \mathbb{E}_{s \sim d_0, a \sim \pi(s)} [\nu(\phi(s,a))], \qquad f_2(\nu) = -\mathbb{E}_{(s,a) \sim d^{\mathcal{D}}} [\nu(\phi(s,a))],$$

$$f_3(\nu) = \mathbb{E}_{(s,a,s') \sim d^{\mathcal{D}}, a' \sim \pi(s')} [\gamma \nu(\phi(s',a'))].$$

Given $n$ samples $\{s_0^{(i)}, a_0^{(i)}, s^{(i)}, a^{(i)}, s'^{(i)}, a'^{(i)}\}_{i=1}^n$ from the generative process $s_0^{(i)} \sim d_0, a_0^{(i)} \sim \pi(s_0^{(i)}), (s^{(i)}, a^{(i)}, s'^{(i)}) \sim d^{\mathcal{D}}, a'^{(i)} \sim \pi(s'^{(i)}) \quad \forall i$, we define the sample-based estimator $\widehat{\mathrm{IPM}}(d_\phi^\pi, d_\phi^{\mathcal{D}})$:

$$\widehat{\mathrm{IPM}}(d_\phi^\pi, d_\phi^{\mathcal{D}})^2 = \frac{1}{n^2} \sum_{i,j} \left[ k(\phi(s^{(i)}, a^{(i)}), \phi(s^{(j)}, a^{(j)})) + (1-\gamma)^2 k(\phi(s_0^{(i)}, a_0^{(i)}), \phi(s_0^{(j)}, a_0^{(j)})) \right.$$

$$+ \gamma^2 k(\phi(s'^{(i)}, a'^{(i)}), \phi(s'^{(j)}, a'^{(j)})) - 2(1-\gamma) k(\phi(s_0^{(i)}, a_0^{(i)}), \phi(s^{(j)}, a^{(j)}))$$

$$\left. - 2\gamma k(\phi(s'^{(i)}, a'^{(i)}), \phi(s^{(j)}, a^{(j)})) + 2\gamma(1-\gamma) k(\phi(s'^{(i)}, a'^{(i)}), \phi(s_0^{(j)}, a_0^{(j)})) \right].$$

By deriving in reverse order, we can recover its another definition in supremum, which can be divided into three parts:

$$\widehat{\mathrm{IPM}}(d_\phi^\pi, d_\phi^{\mathcal{D}}) = \sup_{\nu \in \mathcal{F}} \left| \widehat{f}_1(\nu) + \widehat{f}_2(\nu) + \widehat{f}_3(\nu) \right|$$

$$\text{where} \quad \widehat{f}_1(\nu) = \frac{1-\gamma}{n} \sum_{i=1}^n \nu \left( \phi \left( s_0^{(i)}, a_0^{(i)} \right) \right), \quad \widehat{f}_2(\nu) = -\frac{1}{n} \sum_{i=1}^n \nu \left( \phi \left( s^{(i)}, a^{(i)} \right) \right),$$

$$\widehat{f}_3(\nu) = \frac{\gamma}{n} \sum_{i=1}^n \nu \left( \phi \left( s'^{(i)}, a'^{(i)} \right) \right).$$

We can bound the error of sample-based estimator with individual errors as:

$$
\begin{aligned}
\left| \mathrm{IPM}_{\mathcal{G}}(d_\phi^\pi, d_\phi^{\mathcal{D}}) - \widehat{\mathrm{IPM}}(d_\phi^\pi, d_\phi^{\mathcal{D}}) \right| &= \left| \sup_{\nu \in \mathcal{F}} |f_1(\nu) + f_2(\nu) + f_3(\nu)| - \sup_{\nu \in \mathcal{F}} \left| \widehat{f}_1(\nu) + \widehat{f}_2(\nu) + \widehat{f}_3(\nu) \right| \right| \\
&\leq \sup_{\nu \in \mathcal{F}} \left| |f_1(\nu) + f_2(\nu) + f_3(\nu)| - \left| \widehat{f}_1(\nu) + \widehat{f}_2(\nu) + \widehat{f}_3(\nu) \right| \right| \\
&\leq \sup_{\nu \in \mathcal{F}} \left| f_1(\nu) + f_2(\nu) + f_3(\nu) - \widehat{f}_1(\nu) - \widehat{f}_2(\nu) - \widehat{f}_3(\nu) \right| \\
&\leq \sup_{\nu \in \mathcal{F}} \left[ \left| f_1(\nu) - \widehat{f}_1(\nu) \right| + \left| f_2(\nu) - \widehat{f}_2(\nu) \right| + \left| f_3(\nu) - \widehat{f}_3(\nu) \right| \right] \\
&\leq \sup_{\nu \in \mathcal{F}} \left| f_1(\nu) - \widehat{f}_1(\nu) \right| + \sup_{\nu \in \mathcal{F}} \left| f_2(\nu) - \widehat{f}_2(\nu) \right| + \sup_{\nu \in \mathcal{F}} \left| f_3(\nu) - \widehat{f}_3(\nu) \right|.
\end{aligned}
$$

We then observe that

$$
\begin{aligned}
\sup_{\nu \in \mathcal{F}} \left| f_1(\nu) - \widehat{f}_1(\nu) \right| = (1 - \gamma) \Big\{ &\mathbb{E}_{\substack{s \sim d_0, a \sim \pi(s) \\ \bar{s} \sim d_0, \bar{a} \sim \pi(s)}} [k(\phi(s,a), \phi(\bar{s}, \bar{a}))] \\
&- \frac{2}{n} \sum_{i=1}^{n} \mathbb{E}_{\bar{s} \sim d_0, \bar{a} \sim \pi(s)} \left[ k\left( \phi\left( s^{(i)}, a^{(i)} \right), \phi(\bar{s}, \bar{a}) \right) \right] \\
&+ \frac{1}{n^2} \sum_{i=1}^{n} \sum_{j=1}^{n} k\left( \phi\left( s^{(i)}, a^{(i)} \right), \phi\left( s^{(j)}, a^{(j)} \right) \right) \Big\}^{1/2},
\end{aligned}
$$

which shows that changing $s^{(i)}, a^{(i)}$ results in changes of $\sup_{\nu \in \mathcal{F}} \left| f_1(\nu) - \widehat{f}_1(\nu) \right|$ in magnitude of at most $2(1 - \gamma)\bar{k}^{1/2}/n$ where $\bar{k} = \sup_{z \in \mathcal{Z}} k(z, z)$. Therefore, by McDiarmid's inequality (Theorem A.1),

$$
\Pr \left\{ \sup_{\nu \in \mathcal{F}} \left| f_1(\nu) - \widehat{f}_1(\nu) \right| - \mathbb{E}_{s^{(i)}, a^{(i)}} \left[ \sup_{\nu \in \mathcal{F}} \left| f_1(\nu) - \widehat{f}_1(\nu) \right| \right] \geq \epsilon \right\} \leq \exp \left( -\frac{n\epsilon^2}{2(1 - \gamma)^2 \bar{k}} \right).
$$

Also,

$$
\begin{aligned}
&\mathbb{E}_{s^{(i)}, a^{(i)}} \left[ \sup_{\nu \in \mathcal{F}} \left| f_1(\nu) - \widehat{f}_1(\nu) \right| \right] \\
&= \frac{1 - \gamma}{n} \mathbb{E}_{s^{(i)}, a^{(i)}} \left[ \sup_{\nu \in \mathcal{F}} \left| \mathbb{E}_{\bar{s}^{(i)}, \bar{a}^{(i)}} \left[ \sum_{i=1}^{n} \nu \left( \phi\left( \bar{s}_0^{(i)}, \bar{a}_0^{(i)} \right) \right) \right] - \sum_{i=1}^{n} \nu \left( \phi\left( s_0^{(i)}, a_0^{(i)} \right) \right) \right| \right] \\
&\leq \frac{1 - \gamma}{n} \mathbb{E}_{s^{(i)}, a^{(i)}, \bar{s}^{(i)}, \bar{a}^{(i)}} \left[ \sup_{\nu \in \mathcal{F}} \left| \sum_{i=1}^{n} \nu \left( \phi\left( \bar{s}_0^{(i)}, \bar{a}_0^{(i)} \right) \right) - \sum_{i=1}^{n} \nu \left( \phi\left( s_0^{(i)}, a_0^{(i)} \right) \right) \right| \right] \\
&= \frac{1 - \gamma}{n} \mathbb{E}_{s^{(i)}, a^{(i)}, \bar{s}^{(i)}, \bar{a}^{(i)}, \sigma^{(i)}} \left[ \sup_{\nu \in \mathcal{F}} \left| \sum_{i=1}^{n} \sigma^{(i)} \left\{ \nu \left( \phi\left( \bar{s}_0^{(i)}, \bar{a}_0^{(i)} \right) \right) - \nu \left( \phi\left( s_0^{(i)}, a_0^{(i)} \right) \right) \right\} \right| \right] \\
&\leq 2(1 - \gamma) \sqrt{\frac{\bar{k}}{n}},
\end{aligned}
$$

where the last inequality is from Lemma A.2. Combining the results, we get

$$
\Pr \left\{ \sup_{\nu \in \mathcal{F}} \left| f_1(\nu) - \widehat{f}_1(\nu) \right| - 2(1 - \gamma) \sqrt{\frac{\bar{k}}{n}} \geq \epsilon \right\} \leq \exp \left( -\frac{n\epsilon^2}{2(1 - \gamma)^2 \bar{k}} \right).
$$

Similarly, we derive bounds for $f_2$ and $f_3$ respectively:

$$
\Pr \left\{ \sup_{\nu \in \mathcal{F}} \left| f_2(\nu) - \widehat{f}_2(\nu) \right| - 2 \sqrt{\frac{\bar{k}}{n}} \geq \epsilon \right\} \leq \exp \left( -\frac{n\epsilon^2}{2\bar{k}} \right),
$$

$$
\Pr \left\{ \sup_{\nu \in \mathcal{F}} \left| f_3(\nu) - \widehat{f}_3(\nu) \right| - 2\gamma \sqrt{\frac{\bar{k}}{n}} \geq \epsilon \right\} \leq \exp \left( -\frac{n\epsilon^2}{2\gamma^2 \bar{k}} \right).
$$

By letting RHS of above bounds to be $\delta/3$ and using union bound, we get, with probability 1-$\delta$, we get

$$\left| \text{IPM}_{\mathcal{G}}(d_\phi^\pi, d_\phi^{\mathcal{D}}) - \widehat{\text{IPM}}(d_\phi^\pi, d_\phi^{\mathcal{D}}) \right| \leq \sqrt{\frac{\bar{k}}{n}} \left( 4 + \sqrt{8 \log \frac{3}{\delta}} \right). \tag{13}$$

$\square$

The relationship between $\mathcal{F}$ and $\mathcal{G}$

$$\mathcal{G} = \left\{ g \in (\mathcal{Z} \to \mathbb{R}) : g(z) = \nu(z) - \gamma \mathbb{E}_{\substack{s' \sim T(\phi^{-1}(z)) \\ a' \sim \pi(s')}} [\nu(\phi(s', a'))], \nu \in (\mathcal{Z} \to \mathbb{R}), \langle \nu, \nu \rangle_{\mathcal{H}_k} \leq 1 \right\}$$

show that when the conditional expectation $\mathbb{E}_{s' \sim T(\phi^{-1}(\cdot)), a' \sim \pi(s')}[\nu(\phi(s', a'))] : \mathcal{Z} \to \mathbb{R}$ is a function in RKHS $\mathcal{H}_k$, $\mathcal{G}$ also becomes a subset of $\mathcal{H}_k$.

Then we can prove the following Theorem.

**Theorem 4.3.** *Given an MDP $M$, its estimate $\widehat{M}$ with a bijective representation function $\phi$, i.e. $(\widehat{T}, \widehat{R}) = \langle \widehat{T}_z \circ \phi, \widehat{R}_z \circ \phi \rangle$, and an RKHS $\mathcal{H}_k \subset (\mathcal{Z} \to \mathbb{R})$ induced by a universal kernel $k$ such that $\sup_{z \in \mathcal{Z}} k(z, z) = \bar{k}$, assume that $f_{\phi, \widehat{R}_z, \widehat{T}_z}(z) = \mathbb{E}_{T, \pi} \left[ \sum_{t=0}^{\infty} \gamma^t \mathcal{E}_{\phi, \widehat{R}_z, \widehat{T}_z}(s_t, a_t) \Big| (s_0, a_0) = \phi^{-1}(z) \right] \in \mathcal{H}_k$ with $B_\phi = \|f_{\phi, \widehat{R}_z, \widehat{T}_z}\|_{\mathcal{H}_k}$ and the loss is bounded by $\bar{\mathcal{E}} = \sup_{s \in \mathcal{S}, a \in \mathcal{A}} \mathcal{E}_{\phi, \widehat{R}_z, \widehat{T}_z}(s, a)$. Let $n$ be the number of data in $\mathcal{D}$. With probability $1 - 2\delta$,*

$$\left| R^\pi - \widehat{R}^\pi \right| \leq \frac{1}{n} \sum_{(s,a) \in \mathcal{D}} \mathcal{E}_{\phi, \widehat{R}_z, \widehat{T}_z}(s, a) + B_\phi \widehat{\text{IPM}}(d_\phi^\pi, d_\phi^{\mathcal{D}}) + \sqrt{\frac{\bar{\mathcal{E}}^2}{2n} \log \frac{1}{\delta}} + B_\phi \sqrt{\frac{\bar{k}}{n}} \left( 4 + \sqrt{8 \log \frac{3}{\delta}} \right).$$

*Proof.* Applying the Hoeffding inequality (Hoeffding, 1963), with probability $1 - \delta$, we get:

$$\mathbb{E}_{(s,a) \sim d^{\mathcal{D}}} \left[ \mathcal{E}_{\phi, \widehat{R}_z, \widehat{T}_z}(s, a) \right] \leq \frac{1}{n} \sum_{(s,a) \sim \mathcal{D}} \mathcal{E}_{\phi, \widehat{R}_z, \widehat{T}_z}(s, a) + \sqrt{\frac{\bar{\mathcal{E}}^2}{2n} \log \frac{1}{\delta}}.$$

By using an union bound with Eq. (13) and substituting terms in Eq. (3), we recover the result. $\square$

# B  EXPERIMENT DETAILS

## B.1  COMPUTING INFRASTRUCTURE

All experiments were conducted on the Google Cloud Platform. Specifically, we used compute-optimized machines (c2-standard-4) that provide 4 vCPUs and 16 GB memory for the evaluation experiment of Section 5.1, and we used high-memory machines (n1-highmem-4), which provide 4 vCPUs and 26GB memory, equipped with an Nvidia Tesla K80 GPU for the RL experiment of Section 5.2.

## B.2  DETAILS OF THE OPE EXPERIMENT

**Task details**  We did not modify CartPole-v0 environment and Acrobot-v1 environment from the original implementation of OpenAI Gym (Brockman et al., 2016) except for the maximum trajectory length. We ran PPO (Schulman et al., 2017) to optimize policies to reach a certain performance level and set them as the target policies for CartPole-v0 and Acrobot-v1. For the HIV simulator, we used the code adapted by Liu et al. (2018b), which is originally from the implementation of RLPy[3]. We modified the environment to have more randomness in the initial state (up to $10\%$ perturbation from the baseline initial state) and to use the reward function that gives the logarithm of original reward values, as the original reward function scales up to $10^{10}$. We used a tree-based fitted q-iteration algorithm implemented by Liu et al. (2018b) to optimize the target policy for the HIV simulator. All the other details are shown in Table 2. We assume that the termination conditions of tasks are known in prior.

Table 2: Task settings of OPE experiments.

|  | CartPole-v0 | Acrobot-v1 | HIV simulator |
|---|---|---|---|
| state space dimension | 4 | 6 | 6 |
| # of actions | 2 | 3 | 4 |
| discount rate $\gamma$ | 0.99 | 0.99 | 0.98 |
| # of trajectories | 200 | 200 | 50 |
| max length of training traj. | 200 | 200 | 200 |
| max length of rollouts for evaluation | 1000 | 1000 | 1000 |
| discounted return of target policy | 89.4 | -56.88 | 803.2 |

**Model and algorithm details**  The model we learn is composed of a representation module and a dynamics module. To be consistent with the experiment settings in Liu et al. (2018b), we use a representation module of a single hidden layer feed-forward network that takes the state as input and outputs representation. We squashed the representation between $(-1, 1)$ using the *tanh* activation function. The dynamics module is also a single hidden layer feed-forward network that takes representation as input and outputs state difference and reward prediction for each action. We use the *swish* activation function (Ramachandran et al., 2017) for the hidden layers of two modules. As a whole, the model can also be seen as a feed-forward network with three hidden layers of varying activation functions, where the output of the second hidden layer is the representation we regularize.

For the purpose of comparison, we minimize the $L2$ distance between the model prediction and the desired outcome from data, which corresponds to using a model of Gaussian predictive distribution with fixed variance. We standardized the inputs and outputs of the neural network and used Adam (Kingma & Ba, 2014) with a learning rate of $3 \times 10^{-4}$ for the optimization. When compared to the similar experiments conducted in Liu et al. (2018b), we used a larger and more expressive model with more optimization steps with a smaller learning rate for a more accurate comparison. While the derivation of the RepB-SDE objective was based on the state-action representation function, we use state representation in this experiment for direct comparison with RepBM, which uses state representation (it can be also understood as using action invariant kernel). We follow the choice

---

[3]`https://github.com/rlpy/rlpy`

of Liu et al. (2018b) and use dot product kernel $k(\phi(s), \phi(\bar{s})) = \phi(s)^\top \phi(\bar{s})$ for the OPE experiment, which is not universal but allows us to avoid search of kernel hyperparameters, such as length-scales.

After the training, we generate another 200 trajectories (50 in case of HIV simulator), and rollout in both true and simulated (based on learned model) environments to evaluate models. We measure the individual MSE, which is

$$\text{Individual MSE} = \mathbb{E}_{s_0 \sim d_0}\left[\left(\mathbb{E}_{\widehat{M},\pi}\left[\sum_{t=0}^{\infty} \gamma^t r_t | s_0\right] - \mathbb{E}_{M,\pi}\left[\sum_{t=0}^{\infty} \gamma^t r_t | s_0\right]\right)^2\right]$$

for measuring the performance of each model. Whole experiment, from sampling data to learning and evaluating the model, is repeated 200 times with different random seeds.

**Choice and effect of hyperparameter $\alpha$**  For choosing hyperparameter $\alpha$ for each algorithm, we searched over $\alpha \in \{0.001, 0.01, 0.1, 1, 10\}$ for each off-policyness $\epsilon$ and for each environment. Chosen $\alpha$s were mainly $\alpha \in \{0.001, 0.01\}$ for CartPole-v0, $\alpha \in \{1, 10\}$ for Acrobot-v1, and $\alpha \in \{0.01, 0.1\}$ for HIV simulator for both algorithms. In general, large $\alpha$ was beneficial when high off-policyness ($\epsilon$) is present and/or the task is hard to generalize. On the right we show the example of effect of varying $\alpha$ in CartPole-v0.

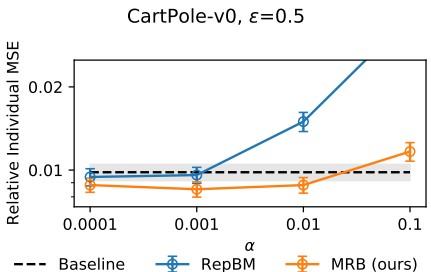

**Comparison to other baselines**  In Figure 2, the OPE results with other model-free baselines are presented. **FQE**: Fitted Q-evaluation, **IS**: step-wise importance sampling, **DR**: doubly robust estimator based on step-wise importance sampling using the value function learned with fitted Q-evaluation (Jiang & Li, 2016), **DualDICE**: stationary distribution correction algorithm (Nachum et al., 2019a). We used the implementation provided by the authors in case of DualDICE. All results are normalized to set the MSE of model-based baseline to be 1. Here, we used average MSEs instead of individual MSEs since importance sampling estimators are not suitable in computing individual MSEs. The results show that model-based methods are more robust to increasing off-policyness when compared to FQE. The results of model-based methods on Acrobot is relatively worse due to the difficult dynamics of Acrobot environment.

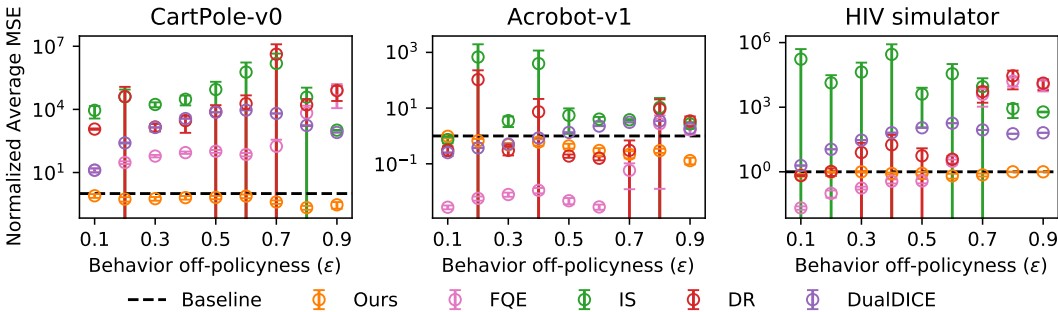

Figure 2: The OPE results compared to other model-free baselines. All experiments are repeated 200 times and the error bars denote 95% confidence interval.

## B.3  DETAILS OF THE OFFLINE RL EXPERIMENTS

**Task details**  We used 12 datasets in D4RL (Fu et al., 2020) over four dataset types and three environments as specified in the main text. In Table 1, normalized scores suggested by (Fu et al., 2020) is used to report the result, where score of 0 corresponds to a random policy and 100 corresponds to a converged SAC policy. In HalfCheetah-v2, score of 0 means the undiscounted return of $-280$, and score of 1 means the undiscounted return of $12135$. In Hopper-v2, score of 0 means the undiscounted return of $-20$, and score of 1 means the undiscounted return of $3234$. In Walker2d-v2 score of 0 means the undiscounted return of $2$, and score of 1 means the undiscounted return of $4592$. We assume that the termination conditions of tasks are known in prior.

**Representation balancing maximizing undiscounted return**  As we report the undiscounted sum of rewards in the experiments, the maximization of lower bound of $R^\pi$ may result in an under-utilization of experiences of later timesteps. One way to mitigate the mismatch is to optimize the policy by maximizing the returns starting from the states in the dataset. It corresponds to maximizing $\tilde{R}^\pi = (1-\gamma)\mathbb{E}_{s_0 \sim d^\mathcal{D}, M, \pi}[\sum_{t=0}^\infty \gamma^t r_t]$ instead of $R^\pi$, where the expectation respect to the initial state distribution $d_0$ is altered with the data distribution $d^\mathcal{D}$.

Consequently, to bound the error of $\tilde{R}^\pi$, the representation should be balanced with another discounted stationary distribution $\tilde{d}^\pi(s,a) \triangleq (1-\gamma)\sum_{t=0}^\infty \gamma^t \Pr(s_t = s, a_t = a | s_0 \sim d^\mathcal{D}, T, \pi)$, the distribution induced by the policy $\pi$ where the initial state is sampled from the data distribution. The derivations can be easily adapted by noting that:

$$\text{IPM}_\mathcal{G}(\tilde{d}^\pi_\phi, d^\mathcal{D}_\phi) = \sup_{\nu \in \mathcal{F}} \left| (1-\gamma)\mathbb{E}_{\substack{s \sim d^\mathcal{D} \\ a \sim \pi(s)}}\Big[\nu\big(\phi(s,a)\big)\Big] - \mathbb{E}_{\substack{(s,a,s') \sim d^\mathcal{D} \\ a' \sim \pi(s')}}\Big[\nu\big(\phi(s,a)\big) - \gamma\nu\big(\phi(s',a')\big)\Big] \right|,$$

and changing the initial state sampling distributions to $d^\mathcal{D}$ during the estimation of IPM.

**Model and algorithm details**  Similar to the model used in the OPE experiment, the model we learn is composed of a representation module and a dynamics module. A representation module is a feed-forward network with two hidden layers that takes the state-action pair as input and outputs representation through the *tanh* activation function. The dynamics module is a single hidden layer network that takes representation as input and outputs parameters of diagonal Gaussian distribution predicting state difference and reward. We use 200 hidden units for all intermediate layers including the representation layer. Across all domains, we train an ensemble of 7 models and pick the best 5 models on their validation error on hold-out set of 1000 transitions in the dataset. The inputs and outputs of the neural network is normalized. We present the pseudo-code of the presented Representation Balancing Offline Model-based RL algorithm below.

---

**Algorithm 1** Representation Balancing Offline Model-based RL

---

**Input**: Offline dataset $\mathcal{D}$, previous policy $\pi$    **Output**: Optimized policy $\pi$

1: Sample $K$ independent datasets with replacement from $\mathcal{D}$.
2: Train bootstrapped ensemble of $K$ models $\{\widehat{T}_i, \widehat{R}_i\}_{i=0}^K$ minimizing Eq. (8) (adapted with $\tilde{d}^\pi_\phi$).
3: **for** $repeat = 0, 1, \ldots$ **do**
4:     **for** $rollout = 0, 1, \ldots, B$ **do**
5:         Sample initial rollout state $s_0$ from $\mathcal{D}$.
6:         **for** $t = 0, 1, \ldots$ **do**
7:             Sample an action $a_t \sim \pi(s_t)$.
8:             Randomly pick $(\widehat{T}_i, \widehat{R}_i)$ and sample $(s_{t+1}, r_t) \sim (\widehat{T}_i, \widehat{R}_i)(s_t, a_t)$.
9:             Compute $\tilde{r}_t = r_t - \gamma\lambda \left\|\sqrt{\mathbb{V}_K[\mu(s_t, a_t)]}\right\|_2$ and store $(s_t, a_t, \tilde{r}_t, s_{t+1})$ in $\widehat{\mathcal{D}}$.
10:         **end for**
11:     **end for**
12:     Draw samples from $\mathcal{D}$ to compute $\widehat{\text{IPM}}(\tilde{d}^\pi_\phi, d^\mathcal{D}_\phi)$.
13:     Draw samples from $\mathcal{D}$ and $\widehat{\mathcal{D}}$ to update critic $Q$.
14:     Maximize $\mathbb{E}_{s \sim \mathcal{D} \cup \widehat{\mathcal{D}}, a \sim \pi(s)}[Q(s,a) - \tau\log\pi(a|s)] - \alpha_\pi\widehat{\text{IPM}}(\tilde{d}^\pi_\phi, d^\mathcal{D}_\phi)$ to update $\pi$.
15: **end for**

---

For the result of **MOPO** (Yu et al., 2020), we ran the code kindly provided by the authors[4] without any modification on the algorithm or the hyperparameters. All algorithms we experimented (**Base**, **RP**, **RepB-SDE**) share all the hyperparameters in common except the ones associated with changing objectives. We run SAC on the full rollouts from the trained ensemble models as shown in Algorithm 1. The common hyperparameters shared among algorithms are shown in Table 3. We simply tried the listed hyperparameters and not tuned them further. For **RP** and **RepB-SDE**, we penalized the reward from simulated environments with the standard deviation of prediction means of neural network ensembles. We used standardized output of all 7 neural networks to compute the reward

---

[4]https://github.com/tianheyu927/mopo

Table 3: Common hyperparameters used in offline RL experiments.

| Parameter | Value |
|---|---|
| optimizer | Adam (Kingma & Ba, 2014) |
| learning rate | $3 \times 10^{-4}$ |
| discount factor $\gamma$ | 0.99 |
| number of samples per minibatch | 256 |
| target smoothing coefficient $\tau$ | $5 \times 10^{-3}$ |
| [actor/critic] number of hidden layers | 2 |
| [actor/critic] number of hidden units per layer | 256 |
| [actor/critic] non-linearity | ReLU |
| # of rollouts | $10^4$ |
| max length of rollouts | $10^3$ |
| rollout buffer size | $5 \times 10^7$ |

Table 4: Normalized scores on D4RL MuJoCo benchmark datasets (Fu et al., 2020) with standard errors fully specified.

| Dataset type | Environment | RepB-SDE (ours) | RP | Base | MOPO |
|---|---|---|---|---|---|
| Random | Walker2d-v2 | $21.1 \pm 1.0$ | $18.4 \pm 1.8$ | $16.4 \pm 2.2$ | $1.3 \pm 0.5$ |
| Medium | Walker2d-v2 | $72.1 \pm 1.9$ | $56.3 \pm 4.3$ | $5.5 \pm 0.4$ | $-0.1 \pm 0.0$ |
| Med-Replay | Walker2d-v2 | $49.8 \pm 11.4$ | $41.1 \pm 10.3$ | $6.2 \pm 0.7$ | $47.8 \pm 4.8$ |
| Med-Expert | Walker2d-v2 | $88.8 \pm 6.9$ | $72.6 \pm 5.1$ | $51.7 \pm 7.1$ | $32.4 \pm 8.8$ |
| Random | Hopper-v2 | $8.6 \pm 1.0$ | $8.3 \pm 0.2$ | $8.3 \pm 0.2$ | $9.1 \pm 0.3$ |
| Medium | Hopper-v2 | $34.0 \pm 2.8$ | $27.5 \pm 3.1$ | $19.8 \pm 0.8$ | $19.2 \pm 4.5$ |
| Med-Replay | Hopper-v2 | $62.2 \pm 6.7$ | $49.8 \pm 8.7$ | $32.9 \pm 5.5$ | $80.8 \pm 2.9$ |
| Med-Expert | Hopper-v2 | $82.6 \pm 7.0$ | $74.0 \pm 7.2$ | $19.1 \pm 1.1$ | $23.2 \pm 1.3$ |
| Random | HalfCheetah-v2 | $32.9 \pm 1.1$ | $31.3 \pm 1.5$ | $26.1 \pm 6.2$ | $29.9 \pm 1.2$ |
| Medium | HalfCheetah-v2 | $49.1 \pm 0.3$ | $47.3 \pm 0.3$ | $22.1 \pm 4.4$ | $45.0 \pm 0.4$ |
| Med-Replay | HalfCheetah-v2 | $57.5 \pm 0.8$ | $53.6 \pm 0.8$ | $54.0 \pm 2.4$ | $53.8 \pm 1.1$ |
| Med-Expert | HalfCheetah-v2 | $55.4 \pm 8.3$ | $36.7 \pm 12.0$ | $31.9 \pm 10.5$ | $91.0 \pm 2.5$ |

penalty. We first search over the reward penalty coefficient $\lambda \in \{0, 2, 5, 7, 10, 15\}$ that grants **RP** the best performance. We shared same $\lambda$ for **RepB-SDE** and searched over $\alpha_M \in \{0.01, 0.1, 1\}$, $\alpha_\pi \in \{0, 0.01, 0.1, 1, 10\}$. We ran the algorithms for $1.5 \times 10^6$ gradient updates, except for HalfCheetah-Medium-Expert where we ran for $5.0 \times 10^6$. In Table 4, we present the standard errors of results, which was omitted in the main text.

In addition to the performance after the final iteration presented in Table 1, we also present the learning curves of the experiment in Figure 3, and the effect of the choice of $\alpha_M$ and $\alpha_\pi$ in Figure 4. The Figure 4 shows that the presented regularization is robust to the choice of $\alpha_M, \alpha_\pi$ except for too large $\alpha_M$, consistently improving from **RP**, which corresponds to $\alpha_M = 0, \alpha_\pi = 0$.

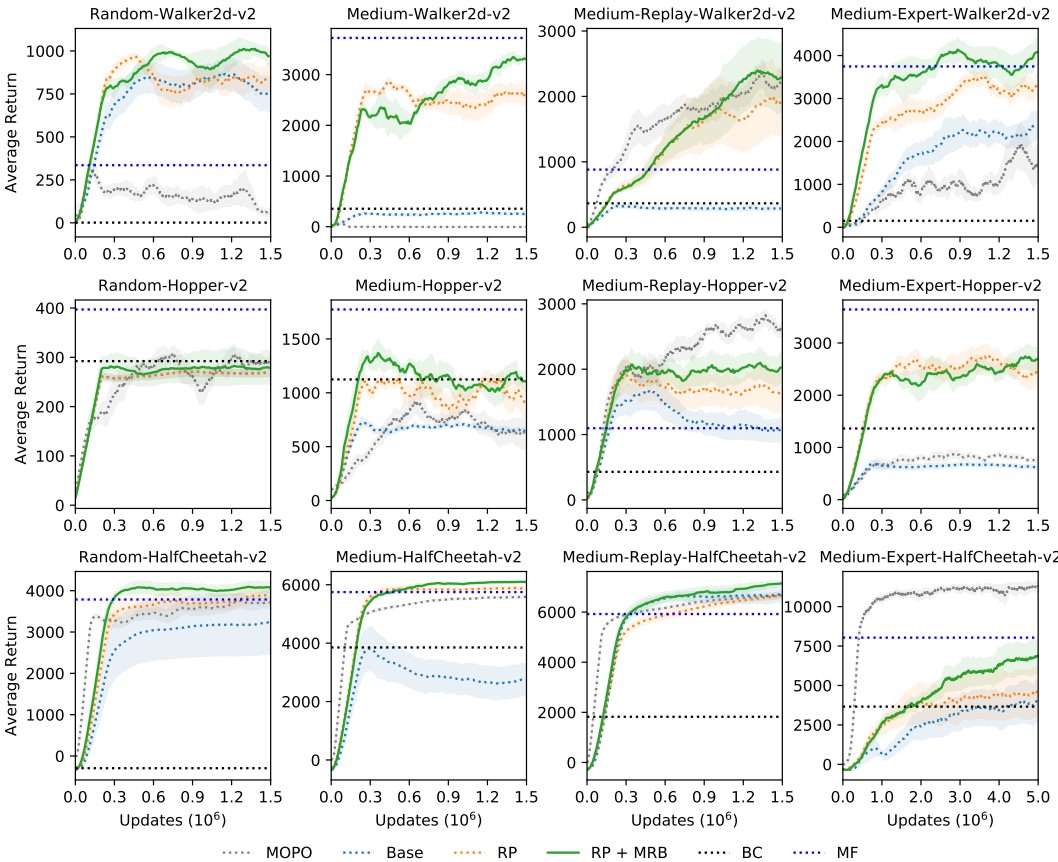

Figure 3: The learning curves of the D4RL experiment.

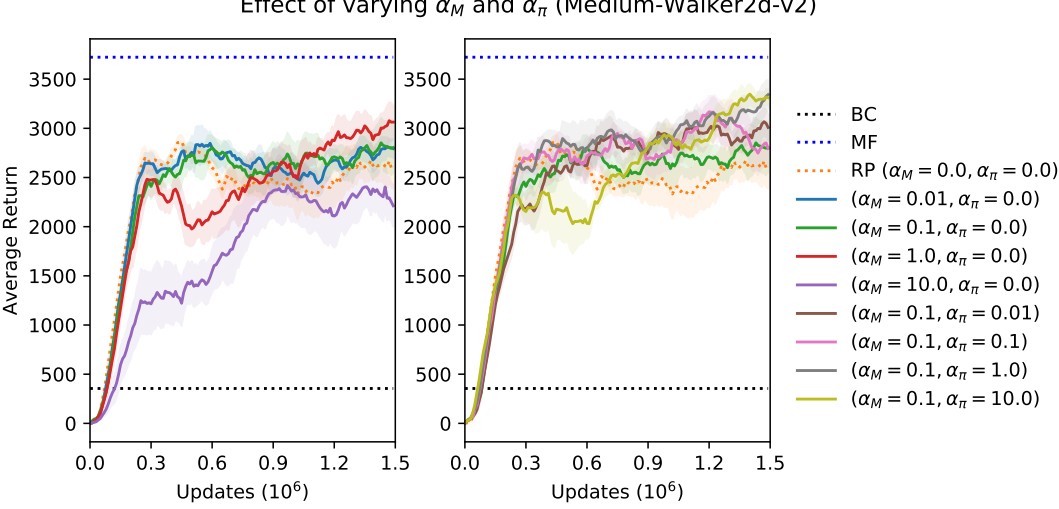

Figure 4: The effect of varying hyperparameter in the D4RL experiment.

