# OpenReview forum: "Representation Balancing Offline Model-based Reinforcement Learning"
_ICLR.cc/2021/Conference — ICLR 2021 Poster_

### Official Review · AnonReviewer1 · 2020-10-18
**Assumptions need better motivation**

**Rating:** 6
**Confidence:** 4

**Review:**

Strength:
Model learning is an important component for offline RL, which is usually done independently from policy evaluation / optimization. The authors propose a new model learning method for offline RL that takes policy evaluation error into consideration / as regularization.
An upper bound is derived to guarantee the worst case performance. In terms of policy evaluation, the authors show empirical advantages over previous model-based offline OPE algorithms. In terms of control, the authors show empirical advantages over existing model-based and model-free algorithms in the challenging D4RL dataset.

Weakness & Points to be clarified:
My major concern is the assumption used in the paper.
The assumption about B_\phi in Theorem 4.3 looks not well motivated. When should we expect there exists such a B_\phi? How large are B_\phi and \bar{k}? If B_\phi and \bar{k} are very large, I feel the bound in theorem 4.3 can be very loose. I think the paper may benefit from clarifying more on this assumption.

Minor comments:
The authors compare with both model-based and model-free approaches in the control setting.
In OPE, however, only model-based approach is compared. I would suggest to add more model-free baselines, e.g., Fitted-Q-Evaluation [1] and DICEs, to motivate the necessity for learning a model.

Overall I think the empirical results are convincing and I am happy to increase my score if the assumptions are further clarified.

[1] Voloshin, Cameron, et al. "Empirical Study of Off-Policy Policy Evaluation for Reinforcement Learning." arXiv preprint arXiv:1911.06854 (2019).

===================

(Nov 24) The authors addressed my concerns in the reply so I increased my score to 6.

---

> ### Author Response · Authors · 2020-11-17
> **About the assumptions in Theorem 4.3**
>
> We sincerely appreciate your valuable comments and feedback.
>
> **Assumptions in Theorem 4.3:** For the ease of notation, in the following comment, we define the expected model loss over trajectories starting from the state represented by $z$,
> $f_{\phi,R,T}(z)=\mathbb{E} _ {T, \pi} [\sum_{t=0}^\infty \gamma^t \mathcal{E}_{\phi,R,T}(s_t,a_t)|(s_0,a_0)=\phi^{-1}(z)]$. We would like to clarify that the assumption on $B_\phi$ in Theorem 4.3 is equivalent to saying that $ f _ { \phi, R, T } (z) \in \mathcal{H} _ k $ with $ B _ \phi = \lVert f _ { \phi, R, T} \rVert _ { \mathcal{H}_k } $, which captures the magnitude and the smoothness of $f _ {\phi, R, T}$ [1]. In general, assuming the underlying dynamics is smooth, we can expect $B_\phi$ to be small when the model error is small. Also, $\bar{k}$ depends on the kernel function we use, but w.l.o.g. we can let $ \bar { k } = 1$and subsume it into $ B _ \phi $, i.e. using $ \tilde { B } _ { \phi } \triangleq B _ \phi \sqrt{ \bar { k } }$. Note that these assumptions are also frequently made in related literature on learning generalizable representations [2, 3, 4].
>
> **Additional baselines of OPE experiment:** We added OPE results with more baselines (IS, DR, FQE, DualDICE) in the Appendix (Figure 2).
>
> [1] Kanagawa, Motonobu, et al. "Gaussian processes and kernel methods: A review on connections and equivalences." arXiv preprint arXiv:1807.02582 (2018).
>
> [2] Shalit, Uri, Fredrik D. Johansson, and David Sontag. "Estimating individual treatment effect: generalization bounds and algorithms." International Conference on Machine Learning. PMLR, 2017.
>
> [3] Liu, Yao, et al. "Representation balancing MDPs for off-policy policy evaluation." Advances in Neural Information Processing Systems. 2018.
>
> [4] Johansson, Fredrik D., David Sontag, and Rajesh Ranganath. "Support and Invertibility in Domain-Invariant Representations." The 22nd International Conference on Artificial Intelligence and Statistics. 2019.

---

> > ### Comment · AnonReviewer1 · 2020-11-24
> > **Response to Authors**
> >
> > Thanks for the updates. I appreciate the clarification about the assumption and raised my score accordingly.

---

### Official Review · AnonReviewer4 · 2020-10-23
**dense presentation gives the reader a hard time**

**Rating:** 7
**Confidence:** 4

**Review:**

Summary:
The paper deals with batch RL (aka offline RL). It builds on "Representation Balancing MDP (RepBM)" and tries to improve the procedure by "stationary DIstribution Correction Estimation (DICE)". DualDICE", a further development of DICE, is used for this purpose.
These procedures are known in each case.
Main claims:
„We empirically show that the model trained by the RepB-SDE objective is robust to the distribution shift for the OPE task, particularly when the difference between the target and the behavior is large.„
The introduced „model-based offline RL algorithm based on RepB-SDE“ has „state-of-the-art performance in a representative set of tasks“ from D4RL.

Strong points:
The paper makes no exaggerated claims.
The treatment of the newer, related works is very good.
The experiments are extensive.
I like the formulation „behavior-agnostic setting where we do not have any knowledge of the data collection
process.“  This expresses the, in my opinion, correct view of the real situation well, while the assumption that there is a "behavior policy" that generated the data is not true in general. It may have been different people at different times who performed the actions while the data set was recorded.

Weak points:
The work is on the one hand very specialized, on the other hand just an incremental modification of existing methods.
The presentation is very dense and quite hard to grasp, even with the Appendix.

Recommendation:
Borderline accept.

Questions:
In the Appendix a potential limitation to deterministic environments is mentioned. Is this just a special case, or is this a true limitation? If it is a true limitation, then this should be mentioned not only in the Appendix, but in the main text.
Why are the uncertainties (standard error) of the other methods not also given in Table 1?

Additional feedback with the aim to improve the paper:
Please be more explicit to make the text more understandable. It should be clarified early that the procedure is applicable to continuous state spaces and continuous action spaces---the initial consideration of the stationary distribution uses discrete state and discrete action spaces. Accordingly the spaces S, A, Z should be defined.

Please correct missing capital letters in the bibliography, e.g. mdps, Algaedice, gaussian
And fix "\phi-divergences" -> "$\phi$-divergences"

"16GB" -> "16 GB"

In Figure 1 the measurement points are connected by lines. Actually lines in such a plot are reserved for a fit to the points or a theoretical curve. The plot would therefore look more scientific if the points were not connected.


===================

(Nov 24)  I increased my score to 7.

---

> ### Author Response · Authors · 2020-11-17
> **Thank you for your constructive comments and feedback.**
>
> We greatly appreciate your thoughtful comments and feedback.
>
> While we build on an existing method (RepBM), we emphasize that the proposed method effectively addresses the significant drawbacks of RepBM, including the curse of horizon and various limitations such as discrete action space and deterministic target policy. Consequently, RepB-SDE can be applied to a much broader set of tasks, e.g. to solve continuous control tasks with deep model-based RL algorithms.
>
> **Deterministic environment:** Our method can be applied to both deterministic/stochastic environments. We used a deterministic environment and L2 distance minimization for the sake of comparison with RepBM, and L2 distance minimization is a special case of our objective. Sorry for the confusion and we have clarified the text in the Appendix.
>
> **Missing standard error:** We could not include the standard errors of other methods in Table 1 due to the space limitation. We included another table (Table 4) with standard errors fully specified in the Appendix.
>
> **Definitions of spaces and the stationary distribution:** We clarified that the framework is applicable to continuous state-action spaces and defined the spaces accordingly. Although the definition of discounted stationary distribution provided is not formal for continuous state-action space, we decided to keep it as is since it is the definition widely adopted in RL literature [1, 2].
>
> **Additional feedback:** We corrected the bibliography, the Appendix, and Figure 1 as suggested.
>
> [1] Nachum, Ofir, et al. "DualDICE: Behavior-agnostic estimation of discounted stationary distribution corrections." Advances in Neural Information Processing Systems. 2019.
>
> [2] Zhang, Ruiyi, et al. "GenDICE: Generalized Offline Estimation of Stationary Values." International Conference on Learning Representations. 2020.

---

> > ### Comment · AnonReviewer4 · 2020-11-24
> > **Response to Authors**
> >
> > I am satisfied with the feedback and the changes to the paper. I raised my rating.

---

### Official Review · AnonReviewer2 · 2020-10-28
**An interesting model-based offline RL**

**Rating:** 7
**Confidence:** 4

**Review:**

In this paper, the authors propose a model-based approach with representation balancing (RepB-SDE)to cope with the distribution shift of offline reinforcement learning. RepB-SDE learns a robust representation for the model learning process, which regularizes the distance between the data distribution and the discount stationary distribution of the target policy in the representation space. RepB-SDE adopts the estimation techniques of DualDICE and a novel point is that RepB-SDE plugs this trick into the model-based representation learning and proposes an effective model-based offline RL algorithm.

The combination of policy estimation techniques and model representation learningin this paper looks interesting and its theoretical derivation is sound. The experiments demonstrate the effectiveness of RepB-SDE over almost 10 baselines in the popular offline RL benchmark D4RL. But this experiment part can be improved, e.g., including the state-of-the-art offline model-free baseline (Kumar et al., 2020; CQL).

This paper is well written, especially its methodology and experiment parts are clear. But I have some concerns about the motivation of RepB-SDE in the introduction part, where this paper says that "However, recent offline RL studies mainly focus on how to improve the policy conservatively while using a common policy evaluation technique without much considerations for the distribution shift." BCQ (Fujimoto et al., 2019) considers the Q-learning (value-iteration)framework of abstracted MDP induced by the given dataset and uses an ensemble of action samplings to deal with continuous action space. From the Q-learning perspective, BCQ can realize effective policy evaluation in the tabular case or with rich expressiveness of Q function. The policy evaluation technique of BCQ does not need importance sampling (see (Haarnoja et al, 2018; SAC) similarly) orDualDICE-like techniques. In order to better presentation, the introduction part needs to be well-motivated and justify recent offline RL algorithms fairly.

---

> ### Author Response · Authors · 2020-11-17
> **About the motivation we made**
>
> Thank you for your constructive comments and feedback.
>
> **Missing baseline:** We included the suggested missing baseline (CQL) in the comparisons. CQL performs the best among model-free algorithms in more than half of the datasets, and by including CQL we found that the best of Model-free methods (MF in Table 1) becomes better than our RepB-SDE in two more datasets. Still, RepB-SDE shows the best average performance in 4 of 12 datasets (same with CQL) and performed competitively in many other datasets. This offline RL performance could be further improved by replacing the simple SAC-based optimization using penalized reward with a more sophisticated model-based RL algorithm.
>
> **Concern about motivation:** The point we aim to address in the introduction was that, in finite MDPs, the fixed point admitted by Q-learning algorithm based on a pre-collected dataset is equivalent to the Q-function of MLE MDP constructed from the corresponding dataset (e.g. see Theorem 3 of SPIBB [1]). Even in the model-free cases, it might be helpful to learn a Q-function that is equivalent to that of some other MDP which is more robust in OPE than simple MLE MDP, unlike simple MSTDE minimization in BCQ. We revised the paper to clarify our point.
>
> [1] Laroche, Romain, Paul Trichelair, and Remi Tachet Des Combes. "Safe policy improvement with baseline bootstrapping." International Conference on Machine Learning. PMLR, 2019.

---

### Official Review · AnonReviewer5 · 2020-11-06
**This paper studies offline model-based RL, proposes a new framework to learn model representation that is robust under distribution shift.**

**Rating:** 7
**Confidence:** 4

**Review:**

- Clarity and Originality:
This paper is well-written and easy to read. The motivation is clearly stated: the original paper [Liu, et al., 2018] highly relies on the marginal action probability ratios to calculate the IPM metric, which suffers the curse of horizon issue. This paper addresses this by utilizing the discounted stationary distribution, which could be estimated using a similar DualDice trick. Theoretically the paper shows an upper bound for the policy evaluation error, which is a balance of model fitting error and the distance between the stationary distributions under behavior policy and target policy. This serves as a reasonable loss function to train the model. Furthermore, the learned model is also being used in learning task through iterative procedures. Empirically, the proposed method compared with the original Rep-BM in OPE setting, and various offline learning method in learning setting.

- Significance:
The paper studies offline RL, which is an important topic in high risk domains. Compared with the existing works, this paper gives a tractable method to explicitly learn the model representation w.r.t the stationary distributions of two policies. This method is pretty general and could be paired with other pessimistic model-based RL methods.

-  Questions:
1. For the evaluation task, it seems the only baseline is RepBM, how's the method compared with other off-policy evaluation methods, such as IS, DR?
2. Typically how do you choose the hyper-parameter $\alpha$, which controls the balance of model fitting and invariance in model representation learning?
3. Compared with Rep-BM, this work seems to upper bound $|R(\pi)-\hat{R}(\pi)|$, instead of MSE, which also results in different loss functions in training M, one is L2 loss, one is log-likelihood, how does this affect the empirical results?

---

> ### Author Response · Authors · 2020-11-17
> **We appreciate your feedback.**
>
> Thank you for your helpful comments and feedback.
>
> **Additional baselines of OPE experiment:** We added OPE results with more baselines (IS, DR, FQE, DualDICE) in the Appendix (Figure 2).
>
> **Choosing the hyper-parameter:** Similar to the other concurrent works on offline RL [1], we ran the algorithm with few different hyperparameters until the convergence and reported the result with the best mean performance. The set of hyperparameters we evaluated and the effect of varying hyperparameters are shown in the Appendix.
>
> **Different loss functions:** Empirically L2 loss is a special case of log-likelihood as we can derive L2 loss by assuming a Gaussian transition model with fixed variance. In OPE experiments where we aim to compare our framework against RepBM, we used L2 loss for our loss function for consistency. In more complex domains, e.g. stochastic environments, assuming transition models with more flexible distributions and training with log-likelihood (i.e. learning the variance parameter as well as the mean parameter) can be better in OPE performance than simply using L2 loss with deterministic transition models.
>
> [1] Kidambi, Rahul, et al. "MOReL: Model-Based Offline Reinforcement Learning." Advances in Neural Information Processing Systems. 2020.

---

### Comment · AnonReviewer4 · 2020-11-10
**D4RL not yet published**

It should be considered, that D4RL is currently under review at ICLR2021 (see https://openreview.net/forum?id=px0-N3_KjA) and not yet published.

---

### Author Response · Authors · 2020-11-24
**Summary of the responses and revisions**

We thank all the reviewers for their time, comments and for providing constructive suggestions. Our responses to the questions and the improvements in the revised paper are summarized below.

**Responses to the questions:**

1. **[R1] Assumptions in Theorem 4.3 about $B _ { \phi } , \bar { k }$ :**  in our case, $B _ { \phi }$ can be seen as an RKHS norm of discounted sum of model errors, and $\bar { k }$ can be subsumed into $B _ \phi$.
2. **[R4, R5] Relation to the loss function of RepBM:** L2 distance minimization of RepBM is a special case of log-likelihood maximization we used. We used L2 distance minimization in OPE experiments for the sake of comparison with RepBM.
3. **[R2] Concern about the motivation:** we referred to BCQ in the introduction because simple Q-learning on a pre-collected dataset converges to the Q-function of MLE MDP in tabular case.
4. **[R5] Hyperparameter selection:** similar to the other concurrent works on offline RL, we reported the result of hyperparameters with the best mean performance.

**Major paper improvements:**
1. We added OPE results with more baselines (IS, DR, FQE, DualDICE) in Appendix B (Figure 2), as suggested by Reviewer 1 and Reviewer 5.
2. We added D4RL benchmark results with standard errors fully specified in Appendix B (Table 4), as suggested by Reviewer 4.
3. We added state-of-the-art baseline (CQL) to the D4RL experiment section, as suggested by Reviewer 2.

**Minor paper improvements:**
1. We corrected typos in the bibliography and the Appendix as suggested by Reviewer 4.
2. We clarified the reason why we used L2 distance minimization in the OPE experiments in Appendix B to address the concern of Reviewer 4.
3. In Section 3, we clarified that we use continuous state spaces and both (discrete and continuous) action spaces throughout the paper, as suggested by Reviewer 4.
4. We removed the line connecting the measurement points in Figure 1, as suggested by Reviewer 4.
5. We clarified the motivation part in the introduction (3rd paragraph) to address the concern of Reviewer 2.

---

### Comment · ~Yue_Wang15 · 2021-03-08
**Open source code**

Hi, all,

Thank your for your great work. I am interested in the proposed method.

Would you please share your code that can reproduce the proposed algorithm?

Thank you very much!

---

> ### Comment · ~Byung-Jun_Lee2 · 2021-03-10
> **About the codebase**
>
> Hi Yue,
>
> Thank you for your interest. The code to reproduce the proposed algorithm is now made public,
>
> and can be found in: (https://github.com/dlqudwns/repb-sde).
>
> Thanks.

---

### Decision · Program_Chairs · 2021-01-07
**Final Decision**

**Decision:**

Accept (Poster)

**Comment:**

The paper studies offline RL, which is an important topic in high risk domains. Compared with the existing works, this paper gives a tractable method to explicitly learn the model representation w.r.t the stationary distributions of two policies. This method is pretty general and could be paired with other pessimistic model-based RL methods.

The experiments are limited to simpler domains, and could be extended to include harder tasks from other continuous control domains. Some examples could be domains such as in Robosuite (http://robosuite.ai/) or Robogym (https://github.com/openai/robogym). These environments have higher dimensional systems with clearer implications of representation learning.

There are concerns on writing style and comprehension.
- The work is on the one hand very specialized, on the other hand just an incremental modification of existing methods.
- The presentation is very dense and quite hard to grasp, even with the Appendix.
- The formalism, while important, can be very loose in terms of bounds. While that does open questions in RL theory, it would be useful for authors to be more candid about this fact in the paper.
I would recommend including the response to R1 in the paper.

Other relevant and concurrent papers to potentially take note of:
- Fine-Tuning Offline Reinforcement Learning with Model-Based Policy Optimization (https://openreview.net/forum?id=wiSgdeJ29ee)
- Robust Offline Reinforcement Learning from Low-Quality Data (https://openreview.net/forum?id=uOjm_xqKEoX)

Given the overall positive reviews, I would recommend acceptance. However, the method would benefit from additional pass on re-writing to make the manuscript more accessible, which in turn to increase impact of this work.